# Modulation of bacterial multicellularity via spatio-specific polysaccharide secretion

Salim T. Islam[ID][1,2,3☯]*, Israel Vergara Alvarez[ID][3☯], Fares Saïdi[1,2,3☯], Annick Guiseppi[3], Evgeny Vinogradov[ID][4], Gaurav Sharma[ID][5,6], Leon Espinosa[ID][3], Castrese Morrone[3], Gael Brasseur[3], Jean-François Guillemot[7], Anaïs Benarouche[8], Jean-Luc Bridot[8], Gokulakrishnan Ravicoularamin[1], Alain Cagna[8], Charles Gauthier[ID][1], Mitchell Singer[ID][5], Henri-Pierre Fierobe[3], Tâm Mignot[ID][3], Emilia M. F. Mauriello[ID][3]*

1 Armand Frappier Health & Biotechnology Research Centre, Institut National de la Recherche Scientifique, Université du Québec, Institut Pasteur International Network, Laval, Québec, Canada, 2 PROTEO, the Quebec Network for Research on Protein Function, Engineering, and Applications, Université Laval, Québec, Québec, Canada, 3 Laboratoire de Chimie Bactérienne, CNRS–Université Aix-Marseille UMR, Institut de Microbiologie de la Méditerranée, Marseille, France, 4 Human Health Therapeutics Portfolio, National Research Council of Canada, Ottawa, Ontario, Canada, 5 Department of Microbiology and Molecular Genetics, University of California–Davis, Davis, California, United States of America, 6 Institute of Bioinformatics and Applied Biotechnology, Electronic City, Bengaluru, Karnataka, India, 7 CNRS–Institut de Microbiologie de la Méditerranée, Marseille, France, 8 Teclis Scientific, Civrieux d'Azergue, France

☯ These authors contributed equally to this work.
* salim.islam@iaf.inrs.ca (STI); emauriello@imm.cnrs.fr (EMFM)

**Data Availability Statement:** All relevant data are within the paper and its Supporting Information files.

## Abstract

The development of multicellularity is a key evolutionary transition allowing for differentiation of physiological functions across a cell population that confers survival benefits; among unicellular bacteria, this can lead to complex developmental behaviors and the formation of higher-order community structures. Herein, we demonstrate that in the social δ-proteobacterium *Myxococcus xanthus*, the secretion of a novel biosurfactant polysaccharide (BPS) is spatially modulated within communities, mediating swarm migration as well as the formation of multicellular swarm biofilms and fruiting bodies. BPS is a type IV pilus (T4P)-inhibited acidic polymer built of randomly acetylated β-linked tetrasaccharide repeats. Both BPS and exopolysaccharide (EPS) are produced by dedicated Wzx/Wzy-dependent polysaccharide-assembly pathways distinct from that responsible for spore-coat assembly. While EPS is preferentially produced at the lower-density swarm periphery, BPS production is favored in the higher-density swarm interior; this is consistent with the former being known to stimulate T4P retraction needed for community expansion and a function for the latter in promoting initial cell dispersal. Together, these data reveal the central role of secreted polysaccharides in the intricate behaviors coordinating bacterial multicellularity.

## Introduction

Multicellularity is denoted by the differentiation of physiological functions across a contiguous cell population, with its development regarded as a key evolutionary transition [1]. To attain

**Funding:** A (i) Discovery operating grant (RGPIN-2016-06637) from the Natural Sciences and Engineering Research Council of Canada (https://www.nserc-crsng.gc.ca/index_eng.asp), (ii) startup grant from the Institut National de la Recherche Scientifique (http://www.inrs.ca/), and (iii) Discovery Award (2018-1400) from the Banting Research Foundation (https://www.bantingresearchfoundation.ca/) fund work in the lab of STI as well as a studentship for FS, who is also a recipient of a graduate studentship from the PROTEO research network (http://proteo.ca/en/). STI was supported by a post-doctoral fellowship (321028) in TM's group at project inception from the Canadian Institutes of Health Research (https://cihr-irsc.gc.ca/e/193.html) and the AMIDEX excellence program of Aix-Marseille University (https://www.univ-amu.fr/en/public/excellence-initiative). Research in the lab of TM is funded by a grant (ANR-15-CE13-0006 BACTOCOMPASS) from the Agence Nationale de la Recherche (ANR) (https://anr.fr/en/), as well as support from the Centre National de la Recherche Scientifique (http://www.cnrs.fr/) and Aix-Marseille University (https://www.univ-amu.fr/fr). Research in the lab of EMFM is supported through a grant from the ANR (ANR-14-CE11-0023-01). IV is supported by a studentship from the CONACYT of Mexico (https://www.conacyt.gob.mx/). Work from the MS lab was supported by a grant (IOS135462) from the National Science Foundation (https://www.nsf.gov/). The funders had no role in study design, data collection and analysis, decision to publish, or preparation of the manuscript.

**Competing interests:** The authors have declared that no competing interests exist.

**Abbreviations:** APE, *Acinetobacter* polyelectrolytic exopolysaccharide; BPS, biosurfactant polysaccharide; BYK, bacterial tyrosine autokinase; D-ManNAc, *N*-acetyl-D-mannosamine; D-ManNAcA, *N*-acetyl-D-mannosaminuronic acid; EPS, exopolysaccharide; ESI-MS, electrospray ionization mass spectrometry; GTase, glycosyltransferase; HSQC, heteronuclear single quantum correlation; IM, inner membrane; LPS, lipopolysaccharide; MASC, major spore coat; NHAc, acetylated amino; NOE, nuclear Overhauser effect; NOESY, nuclear Overhauser effect spectroscopy; OAc, acetoxy; $OD_{600}$, optical density at 600 nm; OM, outer membrane; PCP, polysaccharide co-polymerase; PDB, Protein Data Bank; ppm, parts per million; sfGFP, superfolder green fluorescent protein; ss, signal sequence; T4P, type IV pilus; TMS, transmembrane segment; TOCSY, total correlation spectroscopy; UndPP, undecaprenyl pyrophosphate; WT, wild type.

this level of organizational complexity, cells generally must be able to proliferate, specialize, communicate, interact, and move, with these behaviors promoting an increase in the size of a cell collective and the development of higher-order structures [2]. Though typically associated with metazoan organisms, multicellular physiology is also displayed by bacteria, with the best-studied examples being the formation of biofilms and fruiting bodies [3–6].

Secreted long-chain polysaccharides are an important mediator of multicellularity because they serve to retain and organize cells as well as to physically and biochemically buffer the community within the context of an extracellular matrix [7], thereby enhancing survival and fitness. Monospecies bacterial biofilms have thus been intensively studied with respect to their effects on intercell communication, leading to differences in gene regulation and changes in matrix polysaccharide production. However, in-depth knowledge of the mechanisms used by bacteria to modulate multicellular physiology in such communities is limited.

Because of its complex social predatory lifecycle, the gram-negative δ-proteobacterium *Myxococcus xanthus* has emerged as a leading model system in which to simultaneously study multiple factors contributing to organizational complexity. This soil bacterium is capable of saprophytic feeding on products derived from predation of other bacteria [8]. Two forms of motility are required for this complex physiology: type IV pilus (T4P)-dependent group (i.e., "social" [S]) motility [9,10] on soft surfaces and single-cell gliding (i.e., "adventurous" [A]) motility on hard surfaces mediated by directed transport and substratum coupling of the Agl–Glt trans-envelope complex [11,12]. Upon local nutrient depletion, cells initiate a developmental cycle resulting in aggregation and fruiting body formation within 72 hours, generating 3 differentiated cell subpopulations: (1) cells that form desiccation-resistant myxospores in the center of the fruiting body; (2) those that remain at the base of the fruiting body, termed "peripheral rods"; and (3) forager cells that continue their outward motility away from the fruiting body [13].

*M. xanthus* produces several known long-chain polysaccharides that are central to its complex lifecycle. In addition to the O-antigen polymer that caps its lipopolysaccharide (LPS) and is implicated in motility [14–16], *M. xanthus* biosynthesizes a poorly characterized "slime" polysaccharide that facilitates adhesion of the Glt gliding motility complex proteins to the substratum and is deposited in trails behind surface-gliding cells [17,18]. Exopolysaccharide (EPS) is a specific secreted polymer of this bacterium that is important for T4P-dependent swarm spreading; it is also crucial for biofilm formation because it constitutes a large portion of the extracellular matrix in stationary *M. xanthus* biofilms and connects cells via a network of fibrils [19–21]. The production of EPS requires the presence of a T4P [22], which affects the Dif chemosensory pathway (reviewed elsewhere [13]). Finally, cells undergoing sporulation synthesize the major spore-coat (MASC) polymer that surrounds myxospores [23,24].

The most widespread polysaccharide biosynthesis paradigm is the flippase/polymerase (Wzx/Wzy)-dependent pathway [25]. It is used by gram-negative and gram-positive bacteria as well as Archaea to produce a wide range of secreted and/or cell surface-associated polymers [26] including capsular polysaccharide, adhesive hold-fast polymer, spore-coat polymer, O-antigen, and EPS [27,28]. Wzx/Wzy-dependent polysaccharide assembly is a complex process [29] involving a suite of integral inner-membrane (IM) proteins containing multiple α-helical transmembrane segment (TMS) domains [30]. At the cytoplasmic leaflet of the IM, individual polysaccharide repeat units are built on an undecaprenyl pyrophosphate (UndPP) carrier. UndPP-linked repeats are then translocated across the IM by the Wzx flippase [31,32] via a putative antiport mechanism [33,34]. Defects in this step, resulting in a buildup of UndPP-linked repeat units for a given pathway, can have adverse effects on cell growth as well as polysaccharide synthesis by other pathways in the same cell, all dependent on UndPP-linked sugars [35]. Once in the periplasmic leaflet of the IM, repeat units are joined together by 2 key

periplasmic loops of the Wzy polymerase [36–39], resulting in polymer extension at the reducing terminus of the growing chain [40]. Repeat-polymerization defects in a given pathway requiring UndPP may also affect other pathways requiring UndPP because of sequestration of the cellular UndPP pool. Associated polysaccharide co-polymerase (PCP) proteins determine the modal lengths for the growing polymer; these proteins typically contain 2 TMS tracts with a large intervening coiled-coil periplasmic domain. Wzc proteins of the PCP-2A class (typically from gram-negative species) further contain a cytosolic bacterial tyrosine autokinase (BYK) domain fused to the second TMS of the PCP, whereas PCP-2B Wzc proteins (largely from gram-positive species) are phosphorylated by a separately-encoded Wze tyrosine kinase [41]. For gram-negative bacteria, once a polymer has been synthesized in the periplasm, it is then secreted outside the cell through the periplasm-spanning Wza translocon embedded in the outer membrane (OM) [42,43] (**Fig 1A**). Wzc proteins have also been implicated in polymer secretion likely through contacts formed with their cognate Wza translocon [41].

The *M. xanthus* genome encodes proteins that constitute multiple, yet incompletely annotated, Wzx/Wzy-dependent polysaccharide-assembly pathways. The first of these pathways is responsible for EPS biosynthesis [44,45], whereas the second synthesizes the MASC polymer that surrounds myxospores [46–49]. Disparate Wzx/Wzy-dependent pathway proteins seemingly not involved in either EPS or MASC biosynthesis have also each been studied in isolation [17,46,50], hinting at the possibility of a third such assembly pathway in *M. xanthus* for which the product, as well as the importance of its interplay with other polymers, are entirely unknown.

Herein, we describe the constituents of a newly identified Wzx/Wzy-dependent polysaccharide-assembly pathway in *M. xanthus*, as well as previously unknown [44–49] EPS- and MASC-pathway components; this new third pathway is revealed to synthesize a T4P-inhibited novel biosurfactant polysaccharide (BPS) that exerts direct effects on swarm-level *M. xanthus* behaviors. Spatio-specific modulation of EPS and BPS pathways within *M. xanthus* communities is shown to be important for migration and development. This illustrates the importance of differentially regulated polysaccharide production for complex, coordinated multicellular behaviors in bacteria.

## Results

### *M. xanthus* encodes 3 complete Wzx/Wzy-dependent polysaccharide biosynthesis pathways

To identify the core assembly-and-export constituents of each pathway, proteins with Pfam domains attributed to Wzx (domain: PF13440 [Polysacc_synt]), Wzy (domain: PF04932 [Wzy]), Wzc (domain: PF02706 [Wzz]), and Wza (domain: PF02563 [Poly_export]) were first identified in the *M. xanthus* genome [51]. Protein hits and other cluster constituents were further subjected to fold-recognition comparisons against existing protein structures in the Protein Data Bank (PDB). These data, combined with TMS predictions and intraprotein pairwise alignments, were used to annotate the core components of each of the 3 assembly pathways (**Fig 1A** and **S1 Table**). Genes in the EPS assembly pathway and developmental MASC assembly pathway—(re)named according to longstanding convention [52,53] and given suffixes "X" and "S" to denote "exopolysaccharide" and "spore coat", respectively—were detected in clusters, in line with previous reports [44,47] (**S1A and S1B Fig**). Consistent with the high-molecular-weight polysaccharide-assembly function of Wzx/Wzy-dependent pathways in gram-negative and gram-positive bacteria, numerous predicted glycosyltransferase (GTase) proteins were also found encoded in the immediate vicinity of the identified polymer assembly proteins (**Fig 1B** and **S2 Table**). In addition to the EPS and MASC assembly clusters, we further

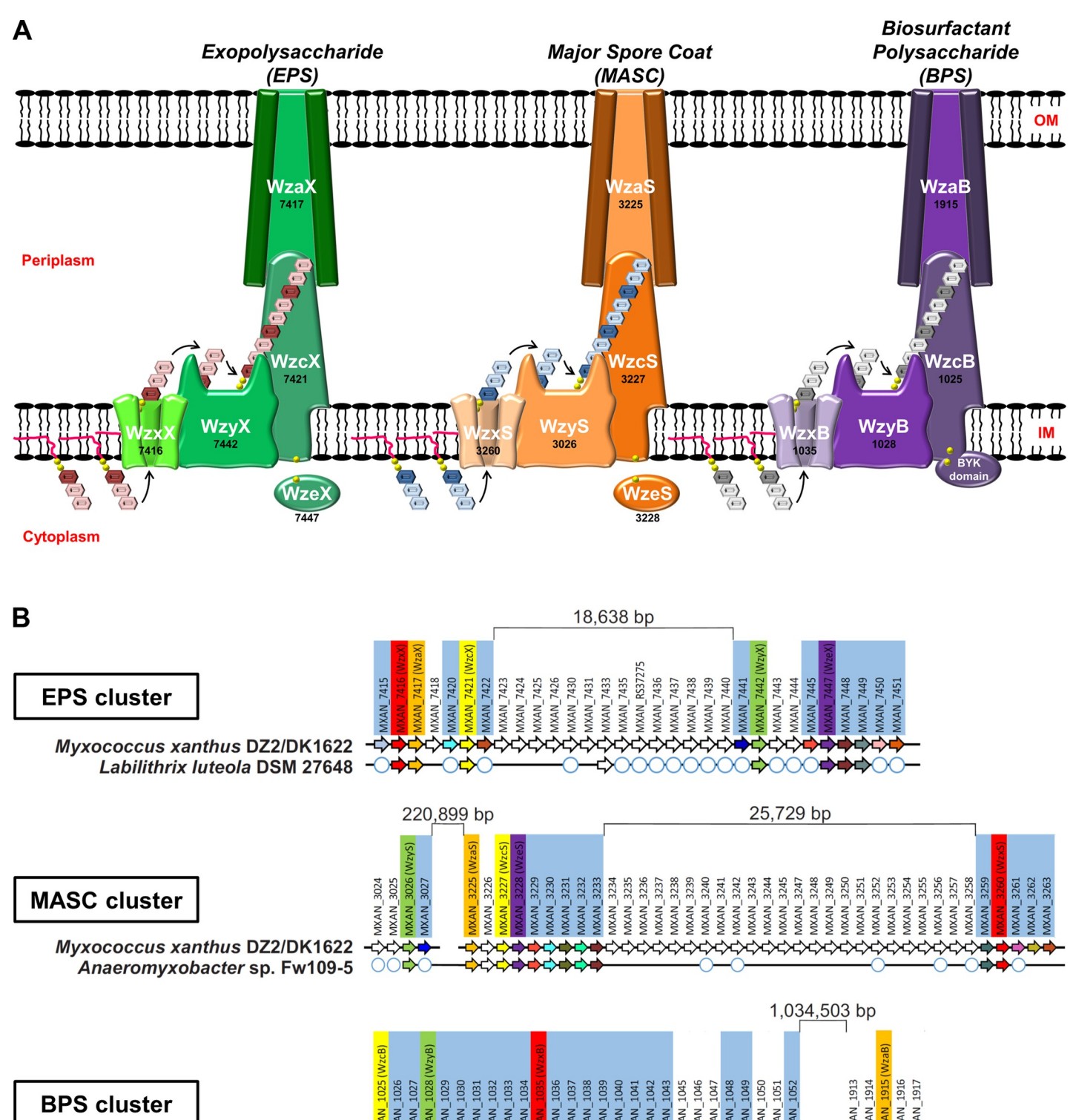

**Fig 1. Wzx/Wzy-dependent polysaccharide biosynthesis pathways encoded by *M. xanthus*. (A)** Schematic representation of the 3 Wzx/Wzy-dependent polysaccharide-assembly pathways in *M. xanthus* DZ2/DK1622. The genomic "MXAN" locus tag identifier for each respective gene (*black*) has been indicated below the specific protein name (*white*). **(B)** Gene conservation and synteny diagrams for EPS, MASC, and BPS clusters in *M. xanthus* DZ2/DK1622 compared to the evolutionarily closest genome containing a contiguous cluster (see **S1 Fig**). Locus tags highlighted by pale blue boxes correspond to genes such as enzymes involved in

monosaccharide synthesis, modification, or incorporation into precursor repeat units of the respective polymer. White circles depict the presence of a homologous gene encoded elsewhere in the chromosome (but not syntenic with the remainder of the EPS/MASC/BPS biosynthesis cluster). BPS, biosurfactant polysaccharide; BYK, bacterial tyrosine autokinase; EPS, exopolysaccharide; IM, inner membrane; MASC, major spore coat; OM, outer membrane.

identified a novel third gene cluster encoding Wzx/Wzy-dependent pathway proteins (given suffixes "B" to denote "biosurfactant", see the section "BPS is a secreted biosurfactant inhibited by the T4P") (S1C Fig). Similar to the EPS and MASC clusters, this third cluster was also highly enriched for genes encoding potential GTases (S2 Table), consistent with this third Wzx/Wzy-dependent pathway also producing a high-molecular-weight polysaccharide (Fig 1A). As Wzx proteins are exquisitely specific to the structure of individual UndPP-linked repeats [31,35], the identification of 3 distinct Wzx flippases (along with their respective GTase-containing gene clusters) is indicative of the production of 3 structurally distinct high-molecular-weight polysaccharides by such assembly pathways in *M. xanthus*.

Though Wzx/Wzy-dependent pathway proteins are typically encoded in contiguous gene clusters [27,54], each of the 3 assembly pathways in *M. xanthus* is encoded at a minimum of 2 separated chromosomal loci. However, homology studies of related genomes revealed syntenic and contiguous clusters in related bacteria, helping to reconcile the various insertions. The *M. xanthus* EPS cluster was found to contain an 18.739-kbp insertion separating the upstream half (encoding WzxX, WzaX, and WzcX) from the downstream half (encoding WzyX and WzeX); however, a contiguous version of this cluster was detected in the genome of *Labilithrix luteola* (Fig 1B and S1A Fig). Even larger insertions were found in the *M. xanthus* MASC cluster, with the tract encoding WzyS separated from the tract encoding WzaS, WzcS, and WzeS by 223.323 kbp; a 25.729-kbp insertion was also identified between the tract encoding WzxS and the WzaS-WzcS-WzeS-encoding segment. Yet, a contiguous MASC assembly cluster lacking the intervening genes was detected in *Anaeromyxobacter* sp. Fw109-5 (Fig 1B and S1B Fig).

Genes encoding assembly proteins WzcB, WzyB, and WzxB were located close to each other on the *M. xanthus* chromosome, interspersed with putative GTase genes, denoting the presence of a potential BPS cluster; however, no gene encoding a WzaB protein to complete this new pathway could be found in close proximity, either upstream or downstream, to the BPS cluster. The only unassigned *wza* gene in the chromosome was the orphan-like *mxan_1915* [17], separated from the BPS cluster by 1.015930 Mbp (Fig 1B and S1C Fig). Remarkably though, homology and synteny studies of related genomes revealed the gene encoding MXAN_1915 (now WzaB) to be contiguous with the genes encoding the other BPS assembly pathway members (i.e., WzcB, WzyB, and WzxB) in the genome of *Sandaracinus amylolyticus* (Fig 1B and S1C Fig), suggesting that MXAN_1915 may be functionally linked with the BPS pathway.

Thus, the EPS, MASC, and BPS clusters all encode putative Wzx, Wzy, Wzc, and Wza proteins, despite the presence of large insertions between certain genes in the *M. xanthus* chromosome. While WzcB from the BPS pathway is a PCP-2A protein (i.e., contains a fused Wze-like C-terminal cytoplasmic BYK domain), both WzcX and WzcS (from the EPS and MASC pathways, respectively) lack such a fusion and are PCP-2B proteins; instead, the EPS and MASC pathways encode stand-alone BYK proteins (WzeX and WzeS, respectively) (Fig 1A).

## EPS and BPS directly affect community organization and behavior

To better understand the functions of the EPS, BPS, and MASC assembly pathways, deletion-mutant strains were constructed for each pathway (S3 Table). In agreement with a previous report [44], all EPS-pathway mutants were severely compromised for swarm expansion and

fruiting body formation (**Fig 2A and 2B**), confirming the central role of EPS in the *M. xanthus* life cycle. Additionally, MASC⁻ cells displayed wild-type (WT)-like group motility and fruiting body formation (**Fig 2A and 2B**), consistent with MASC-pathway expression only during developmental phases [55]. To limit the scope of this project to vegetative cell physiology, minimal additional characterization of the MASC pathway is reported herein.

Cells deleted for BPS-cluster genes displayed "fuzzy" swarm morphology during T4P-dependent swarm spreading (**Fig 2B**). Fruiting body formation in the absence of BPS also required lower cell densities (**Fig 2B** and **S2A Fig**), suggesting that BPS⁻ cells aggregate more easily. This potentially higher aggregative capacity may also help explain why BPS⁻ swarms are compromised for T4P-dependent surface spreading (**Fig 2A and 2B**). Neither BPS nor EPS was required for predation, as mutants defective in either pathway were still able to invade and digest a colony of prey *Escherichia coli* cells (**S2B Fig**) on a hard substratum. Prey invasion is typically followed by the formation of ripples (i.e., synchronized waves of coordinated cells) (**S2B Fig**), a phenomenon that increases the efficiency of prey cell lysis [56]. Although both BPS⁻ and EPS⁻ cells were still able to invade the *E. coli* colony, only BPS⁻ swarms displayed WT-like rippling, whereas EPS⁻ swarms did not ripple (**S2B Fig**). This suggests that (1) EPS may be required for rippling, and if so, (2) BPS⁻ cells may still elaborate cell-surface EPS.

Importantly, the *mxan_1915* (now *wzaB*) mutant displayed identical phenotypes to those of the other BPS-pathway mutants (**Fig 2**), despite the 1.015930-Mbp separation of the gene from the rest of the BPS biosynthesis cluster (**Fig 1B**), further supporting the notion that *mxan_1915* indeed encodes for the BPS-pathway Wza (**Fig 1A**). In support of this hypothesis, a Δ*mxan_1915* Δ*wzcB* double mutant displayed phenotypes similar to those of the respective single mutants, consistent with *mxan_1915* (now *wzaB*) and *wzcB* encoding proteins that belong to the same (i.e., BPS) pathway (**Fig 2A and 2B**).

## BPS is not bound to the cell surface

The (1) hyperaggregative character and (2) rippling phenotype during predation of BPS⁻ cells (**S2 Fig**) suggested the presence of EPS on the surface of these cells. To address these observations, we set out to better understand the nature of the surface-associated polysaccharide layer in BPS⁻ cells. As retention of trypan blue dye has long been used as a readout for production of cell-surface EPS in *M. xanthus* [57], we obtained dye-binding profiles for all EPS- and BPS-pathway mutant strains. For all proposed EPS-pathway mutants, trypan blue binding was drastically reduced compared with WT cells (**Fig 3A**), consistent with previous descriptions of EPS deficiencies [57]. However, BPS-pathway mutants exhibited divergent trypan blue-binding profiles: (1) Mutants unable to flip or polymerize UndPP-linked BPS repeat units (i.e., Δ*wzxB* and Δ*wzyB*) displayed significantly lower dye binding than WT cells (**Fig 3A**), consistent with reduced EPS production in these backgrounds. The likeliest explanation is that mutant strains in which UndPP-linked oligosaccharide repeat units for a particular pathway may build up can manifest polysaccharide synthesis defects in other pathways also requiring UndPP-linked units [58]. (2) Conversely, *M. xanthus* BPS-pathway mutants with the capacity for periplasmic polymerization of the BPS chain but compromised for BPS secretion (i.e., Δ*wzcB*, Δ*wzcB*ᵦᵧₖ, Δ*wzaB*, and Δ*wzaB* Δ*wzcB*) did not display reduced trypan blue binding relative to WT cells (**Fig 3A**). Compared with BPS-pathway Δ*wzaB*, the dye-binding profile of the EPS- and BPS-pathway double mutant Δ*wzaX* Δ*wzaB* matched that of the EPS-pathway Δ*wzaX* single mutant (**Fig 3A**). These genetic results reinforce the notion that the EPS and BPS biosynthesis pathways are independent of each other.

Because the trypan blue-binding assay permits the detection of cell-associated polysaccharides as part of a contiguous surface matrix, the observation that dye binding by BPS⁻ cells was

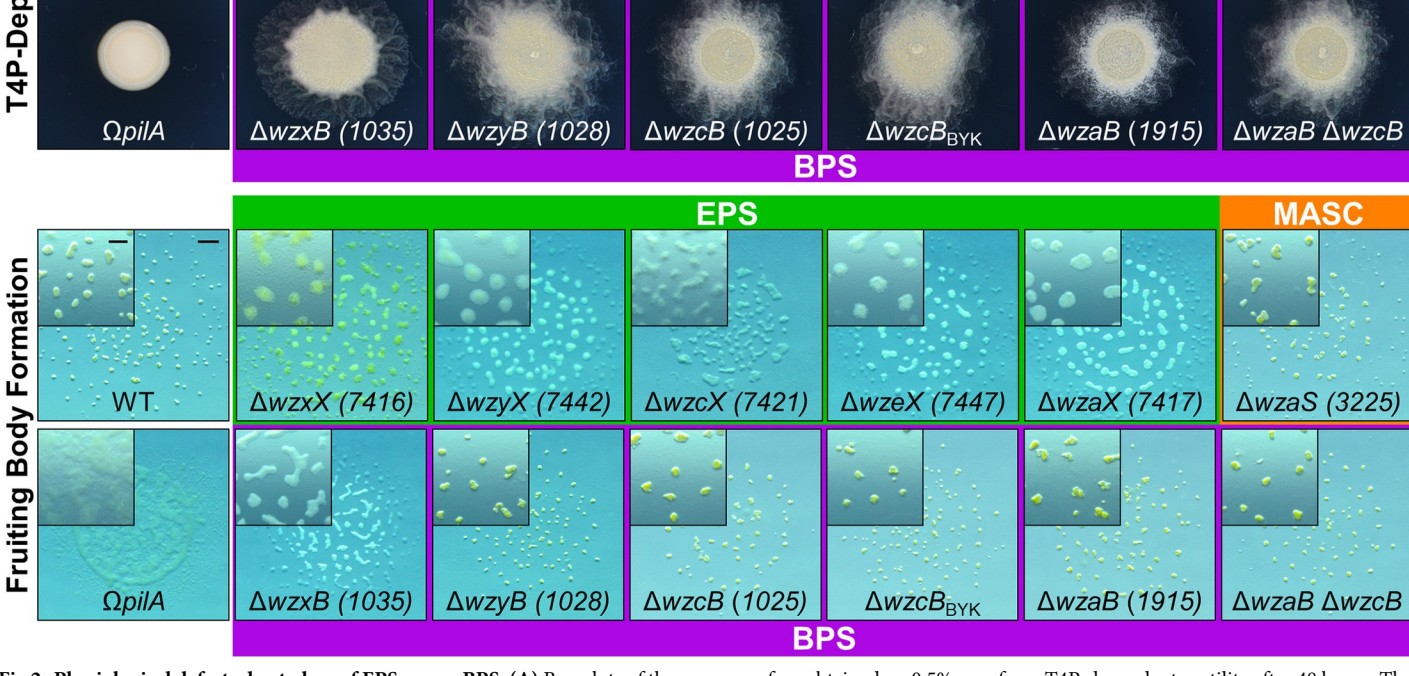

**Fig 2. Physiological defects due to loss of EPS versus BPS. (A)** Box plots of the swarm surface obtained on 0.5% agar from T4P-dependent motility after 48 hours. The lower and upper boundaries of the boxes correspond to the 25th and 75th percentiles, respectively. The median (line through center of boxplot) and mean (+) of each dataset are indicated. Lower and upper whiskers represent the 10th and 90th percentiles, respectively; data points above and below the whiskers are drawn as individual

points. Asterisks denote datasets displaying statistically significant dataset differences ($p < 0.05$) compared with WT, as determined via 1-way ANOVA with Tukey's multiple comparisons test. A minimum of 4 biological replicate values were obtained, each the mean of 3 technical replicates. Raw values and detailed statistical analysis are available (S1 Data). **(B)** EPS-, MASC-, and BPS-pathway mutant swarm physiologies. Top: T4P-dependent motility after 48 hours (scale bar: 2 mm). Bottom: Fruiting body formation after 72 hours (main panel, scale bar: 1 mm; magnified inset, scale bar: 400 μm). BPS, biosurfactant polysaccharide; EPS, exopolysaccharide; MASC, major spore coat; T4P, type IV pilus; WT, wild type.

not reduced compared with WT cells suggests 3 possible scenarios: (1) BPS is surface associated but does not bind trypan blue; (2) BPS is instead secreted to the extracellular milieu; or (3) the BPS-pathway machinery does not produce a polysaccharide and instead has a novel alternative function. The latter possibility is unlikely considering the abundance of putative GTase genes in the BPS cluster (S2 Table). The hypothesis that BPS is not cell associated (and instead likely secreted) is supported by the results of monosaccharide analysis of cell-associated polymers from surface-grown WT and BPS⁻ submerged cultures; these analyses revealed that the major cell-associated sugar species are analogous in both composition and quantity between WT and BPS⁻ strains (S3A Fig), indicating that trypan blue is binding to the same polysaccharide target in each strain. Therefore, because WT and BPS⁻ cells elaborate comparable levels of EPS and do not display differences in surface-associated sugar content, we conclude that BPS does not remain attached to the cell surface.

## BPS is a secreted biosurfactant inhibited by the T4P

Given the (1) aggregative phenotypes of BPS⁻ cells during fruiting body formation (S2A Fig), (2) compromised swarm spreading on surfaces (Fig 2A and 2B), and (3) lack of cell-associated sugar differences (S3A Fig), we reasoned that BPS may be a secreted surface-active biopolymer with emulsifying and/or surfactant properties that should therefore be found in the extracellular environment. We thus employed 2 independent methods to probe the surface-active properties of secreted BPS. The first approach takes advantage of the ability of both emulsifiers and surfactants to stabilize emulsions of 2 immiscible phases [59]. As *pilA* mutations reduce clumping of *M. xanthus* cells in liquid culture because of reduction (not elimination) of EPS production [60] (S3B Fig), these mutant strains can thus be grown to higher overall cell densities than piliated variants; we used this approach to naturally maximize the concentration of secreted polymers in our cultures. Enriched (aqueous) supernatants from dense cultures were thus extracted, filtered, and tested for the ability to stably mix with the hydrocarbon hexadecane [61,62]; light transmission through samples in cuvettes was then monitored in real time via spectrophotometry to monitor emulsion breaking. Whereas BPS⁻ supernatants demonstrated rapid phase separation, supernatants from BPS⁺ cultures formed more stable emulsions with hexadecane (Fig 3B); the latter samples even required 1.6 times more time to simply allow any detectable light to pass through the sample from the start of attempted $OD_{600}$ (optical density at 600 nm) readings until the first registered value (Fig 3B, inset). BPS thus possesses emulsion-stabilizing properties, a feature of both emulsifiers and surfactants.

In addition to increasing the stability of 2 immiscible phases (i.e., possessing emulsifying properties), to be considered a surfactant, a given compound must also be able to reduce the surface/interfacial tension between 2 phases [59]. We thus also probed differences in surface tension between supernatants from submerged WT, EPS⁻, and BPS⁻ swarms using a highly sensitive dynamic drop tensiometer. This method revealed a higher surface tension in BPS⁻ colony supernatants compared with supernatants from WT and EPS⁻ colonies that are still able to secrete BPS (Fig 3C). These data are consistent with surfactant properties of culture supernatants in strains with an intact BPS biosynthesis pathway.

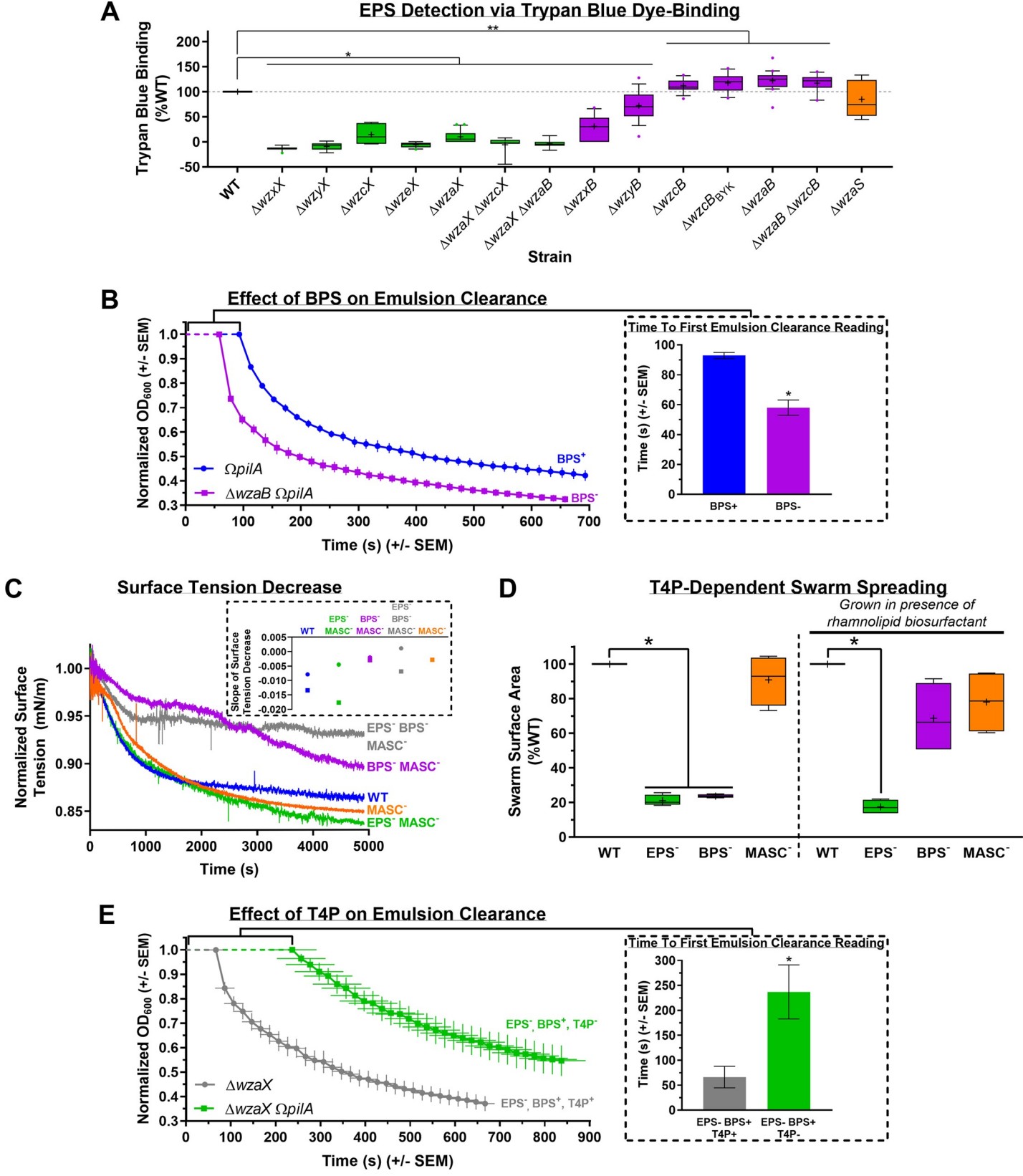

**Fig 3. Analysis of BPS properties. (A)** Boxplots of trypan blue dye retention to indicate the levels of surface-associated polysaccharide production in various strains relative to WT. The lower and upper boundaries of the boxes correspond to the 25th and 75th percentiles, respectively. The median (line through center of boxplot) and mean (+) of each dataset are indicated. Lower and upper whiskers represent the 10th and 90th percentiles, respectively; data points above and below the whiskers are drawn as individual points. Asterisks denote datasets displaying statistically significant differences in distributions ($p < 0.05$) shifted higher (**) or lower (*) than WT, as determined via Wilcoxon signed-rank test performed relative to "100." Raw values and detailed statistical analysis are available (**S2 Data**). **(B)** Real-time clearance of hexadecane–CYE supernatant emulsions from BPS$^+$ and BPS$^-$ strains; values are the mean of 3 biological replicates (+/− SEM). $OD_{600}$ values were normalized to their first registered values, whereas registered time points are displayed at their actual occurrence. Inset: Scanning time (post-mixing) for a given cuvette (containing a culture supernatant–hexadecane emulsion) until a first absolute value for $OD_{600}$ could be registered by the spectrophotometer, for samples with(out) BPS ($n = 3$). Asterisk (*) denotes statistically significant difference in mean value compared with *pilA* mutant ($p = 0.0031$), as determined via unpaired Student's *t* test. Raw values and detailed statistical analysis are available (**S2 Data**). **(C)** Time course of normalized surface tension values (via digital-drop tensiometry) from representative submerged-culture supernatants. Surface tension values across all time points were normalized against the initial surface tension value ($t = 0$) for each respective strain (**S3C Fig**). Strains tested: WT, MASC$^-$ (Δ*wzaS*), BPS$^-$ MASC$^-$ (Δ*wzaB* Δ*wzaS*), EPS$^-$ MASC$^-$ (Δ*wzaX* Δ*wzaS*), EPS$^-$ BPS$^-$ MASC$^-$ (Δ*wzaX* Δ*wzaB* Δ*wzaS*). Inset: Slope values from biological replicate time courses (each represented by a different shape) for each strain. Slopes were calculated by fitting the time-course curves with a fourth-degree polynomial function. Raw values are available (**S3 Data**). **(D)** T4P-dependent swarm spreading in the presence of exogenous di-rhamnolipid-C$_{14}$-C$_{14}$ biosurfactant from *Burkholderia thailandensis* E264. The lower and upper boundaries of the boxes correspond to the 25th and 75th percentiles, respectively. The median (line through center of boxplot) and mean (+) of each dataset are indicated. Lower and upper whiskers represent the 10th and 90th percentiles, respectively. Asterisks denote datasets displaying statistically significant differences in mean values ($p < 0.05$) compared with WT swarms, as determined via 1-sample *t* test performed relative to "100." Raw values and detailed statistical analysis are available (**S1 Data**). **(E)** Real-time clearance of hexadecane–CYE supernatant emulsions from T4P$^+$ and T4P$^-$ BPS-producing strains; values are the mean of 3 biological replicates (+/− SEM). $OD_{600}$ values were normalized to their first registered values, whereas registered time points are displayed at their actual occurrence. Inset: Scanning time (post-mixing) for a given cuvette (containing a culture supernatant–hexadecane emulsion) until a first absolute value for $OD_{600}$ could be registered by the spectrophotometer, for samples with(out) a functional T4P ($n = 4$). Asterisk (*) denotes statistically significant difference in mean value compared with Δ*wzaX* ($p = 0.0265$), as determined via unpaired Student's *t* test. Raw values and detailed statistical analysis are available (**S2 Data**). BPS, biosurfactant polysaccharide; CYE, casitone-yeast extract; EPS, exopolysaccharide; MASC, major spore coat; $OD_{600}$, optical density at 600 nm; T4P, type IV pilus; WT, wild type.

To further test the function of BPS as a biosurfactant, we sought to rescue the motility phenotype of BPS$^-$ swarms through addition of an exogenous biosurfactant, di-rhamnolipid-C$_{14}$-C$_{14}$ produced by *Burkholderia thailandensis* E264 [63]. This biosurfactant is not synthesized via a Wzx/Wzy-dependent polysaccharide-assembly pathway; it is instead produced in the cytoplasm of its native bacterium requiring a 3-gene *rhl* cluster [63]. This renders rhamnolipid chemically distinct from high-molecular-weight biopolymers, while able to reduce surface tension through its strong surfactant properties. As reported, relative to WT swarms, EPS$^-$ or BPS$^-$ strains were severely compromised for T4P-dependent swarm spreading, whereas MASC$^-$ swarms displayed no significant differences (**Fig 2A**). However, upon pretreatment of the agar surface with exogenous rhamnolipid biosurfactant, BPS$^-$ swarms regained near-WT-like spreading (whereas EPS$^-$ and MASC$^-$ swarms displayed the same phenotypes as those in the absence of rhamnolipid) (**Fig 3D** and **S3D Fig**). Exogenous biosurfactant addition is thus able to complement BPS deficiency in trans.

As EPS production has long been known to require the presence of a T4P [9,22] (**S3B Fig**), we used the aforementioned hexadecane-based bioemulsifier assay and high-density cultures to test the T4P-dependence of surface-active BPS production. For supernatants from EPS-deficient BPS$^+$ cultures encoding a T4P, gradually breaking emulsions were observed for samples encoding a functional T4P; however, inactivation of the T4P in this parent strain gave rise to culture supernatants with remarkably extended abilities to stabilize emulsions (**Fig 3E**). From the beginning of attempted $OD_{600}$ readings until the first detected value, the latter samples took 3.6 times as long just to allow any detectable light to pass through the cuvette (**Fig 3E, inset**). Thus, as opposed to EPS production, which requires a T4P [22], the presence of a T4P may have an inhibitory effect on the production of surface-active BPS.

## BPS is a randomly acetylated repeating ManNAcA-rich tetrasaccharide

We thus set out to characterize the structure and composition of the novel secreted BPS molecule. Given the strong surface-active properties in enriched supernatants from BPS$^+$ Δ*wzaX* Ω*pilA* cultures (**Fig 3E**) and lack thereof from BPS$^-$ Δ*wzaB* Ω*pilA* enriched supernatants (**Fig 3B**), we studied the differences in polysaccharide content between these samples. Protein/

nucleic acid-depleted samples were first analyzed via 1D proton NMR, revealing a cluster of peaks in the BPS$^+$ enriched supernatant near 2 parts per million (ppm) that were not present in the BPS$^-$ sample (Fig 4A). Samples were then separated via gel chromatography, revealing the presence of an acidic polysaccharide in the BPS$^+$ sample (S4A Fig). Anion-exchange chromatography followed by $^1$H–$^{13}$C heteronuclear single quantum correlation (HSQC) NMR analysis of the isolated polysaccharide revealed a complicated NMR spectrum due to random acetylation (S4B Fig). However, subsequent HSQC analysis of a deacetylated sample revealed well-defined resonances (Fig 4B and Table 1). In the spectrum of the O-deacetylated polysaccharide, 4 spin systems were observed, each a pyranose sugar with an acetylated amino (NHAc) group at carbon 2, identified by $^{13}$C signal position between 50 and 60 ppm. All sugars displayed β-manno-configuration, as identified via total correlation spectroscopy (TOCSY) and nuclear Overhauser effect spectroscopy (NOESY) signal patterns and agreement with $^{13}$C signal positions (Table 1). One monosaccharide (residue A) contained a CH$_2$OH group at carbon 5 and was thus designated N-acetyl-mannosamine. The remaining 3 sugars (residues B, C, and D) displayed no further correlations from hydrogen 5 and thus were classified as N-acetyl-mannosaminuronic acids, designations that were confirmed by the mass spectrum (Fig 4C). The sequence of the monosaccharides was identified based on nuclear Overhauser effect (NOE) detection between nuclei A1:D3, B1:A4, C1:B4, and D1:C4 (Fig 4B). Because of signal overlap, it was not possible to fully differentiate the latter 2 NOEs (Fig 4B), but this did not affect monosaccharide sequence designation.

Using these data, we were able to revisit the spectrum annotation for the untreated polysaccharide. The original polymer was found to be partially acetylated at position 3 in residues A, B, and C (S4B Fig). For residue D, oxygens 4 and 6 were capable of being acetylated, but no appropriate signal was visible; instead, positions 4 and 6 in residue D yielded the same signals as those detected in the deacetylated polysaccharide (Fig 4B and 4C). For residues A, B, and C, only oxygen 3 was free to be acetylated, thus agreeing with the signal position. Three acetoxy (OAc) signals were present in the HSQC spectrum: 2.02/21.6; 2.04/21.6; 2.10/21.6 ppm; although there may be overlaps (S4B Fig).

Taken together, these analyses revealed BPS to be a heteropolysaccharide built of the following repeating unit: →3)-D-ManNAcA-(β1→4)-D-ManNAcA-(β1→4)-D-ManNAcA-(β1→4)-D-ManNAc-(β1→ (Fig 4D). Each tetrasaccharide repeat is thus joined via β1→3-linkages to another, with each repeating unit composed of a proximal neutral β-configuration N-acetyl-D-mannosamine (D-ManNAc) sugar, followed by 3 distal charged β-configuration N-acetyl-D-mannosaminuronic acid (D-ManNAcA) sugars; a random acetylation pattern at position 3 was detected for the first 3 residues of the tetrasaccharide (Fig 4D). Therefore, the product of the Wzx/Wzy-dependent BPS assembly pathway (Fig 1A) is indeed a high-molecular-weight biosurfactant sugar polymer built of randomly acetylated mannosaminuronic acid-rich tetrasaccharide repeat units (Fig 4D).

## BPS and EPS polymers are shared differently within the swarm community

Given the compromised nature of T4P-dependent swarm spreading for both EPS$^-$ and BPS$^-$ strains (Fig 2), we mixed populations of both strains together to test the ability of each polysaccharide to cross-complement deficiency in the other and restore swarm motility on soft agar. Mixture of the 2 strains at a 1:1 ratio restored T4P-dependent swarm expansion to WT levels (Fig 5A and 5B), indicating an in trans complementation of a motility defect. We next explored the spatial distribution of the 2 strains at a 1:1 ratio via fluorescence microscopy of EPS$^-$ cells with a superfolder green fluorescent protein (sfGFP)-labeled OM and BPS$^-$ cells elaborating an mCherry-labeled IM [18] (Fig 5C). Although both EPS-pathway (green) and

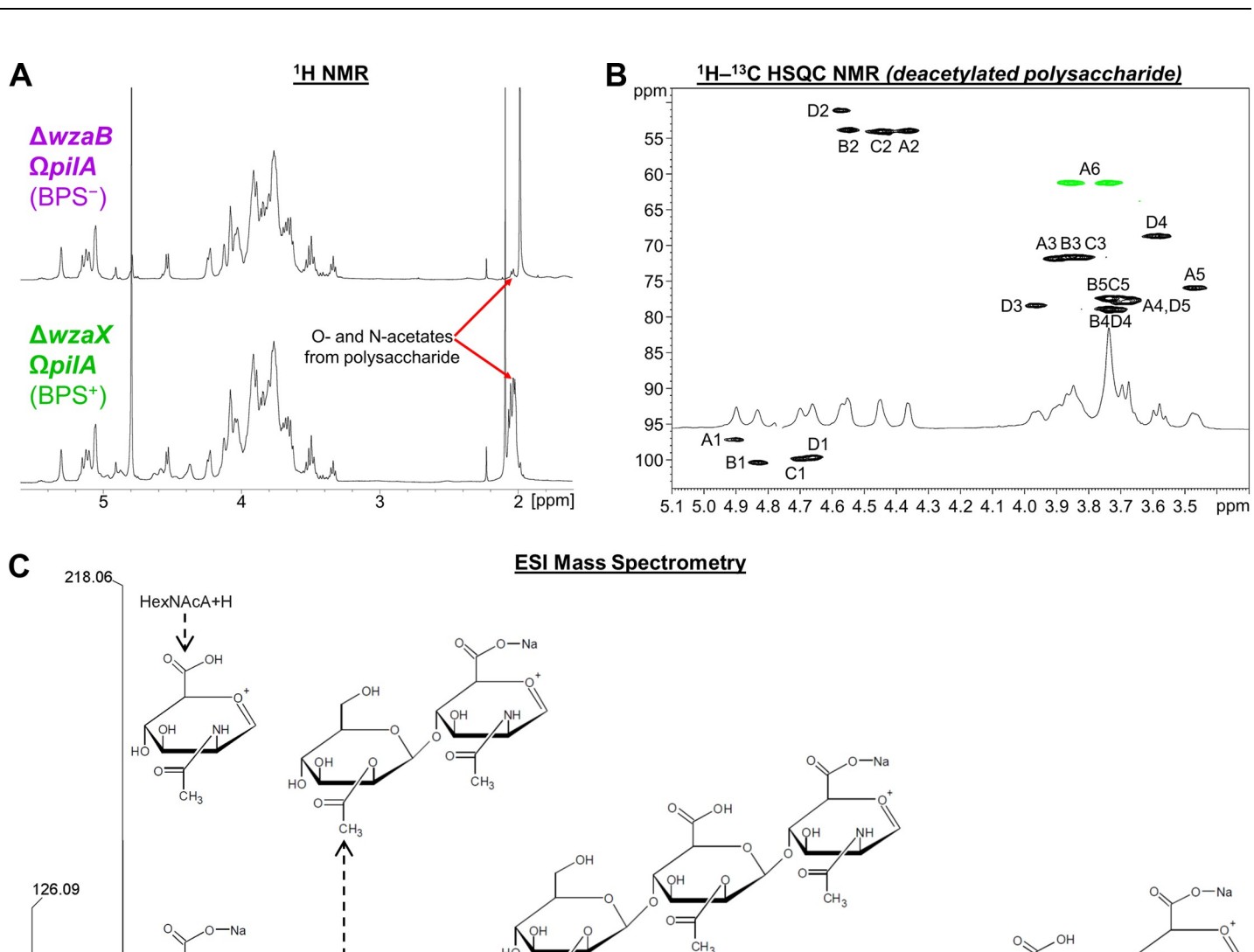

**D** →3)-D-ManNAcA-(β1→4)-D-ManNAcA-(β1→4)-D-ManNAcA-(β1→4)-D-ManNAc-(β1→
　　　　D　　　　　　　　　　　　C　　　　　　　　　　　　B　　　　　　　　　　　A

**Fig 4. Analysis of BPS composition and structure. (A)** [1]H NMR spectra of concentrated supernatants from BPS[+] Δ*wzaX* Ω*pilA* and BPS[−] Δ*wzaB* Ω*pilA* cultures. **(B)** [1]H–[13]C HSQC spectrum of deacetylated acidic polysaccharide originally isolated from Δ*wzaX* Ω*pilA* supernatant. Analysis was performed at 27˚C, 500 MHz. Resonance peak colors: black, C–H; green, C–H$_2$. **(C)** Negative-mode high cone voltage (180 V) ESI-MS of deacetylated acidic polysaccharide from Δ*wzaX* Ω*pilA* supernatant. **(D)** Chemical structure of the BPS polymer tetrasaccharide RU. BPS, biosurfactant polysaccharide; D-ManNAc, *N*-acetyl-D-mannosamine; D-ManNAcA, *N*-acetyl-D-mannosaminuronic acid; ESI-MS, electrospray ionization mass spectrometry; HSQC, heteronuclear single quantum correlation; RU, repeating unit.

BPS-pathway (red) mutant cells were detected in the swarm interior, as well as at its edge, the distribution was not homogeneous (**Fig 5C**); BPS[−] cells (red, i.e., cells still able to produce EPS) were more abundant toward the center of the swarm, whereas EPS[−] cells (green, i.e., cells still able to produce BPS) were enriched toward the periphery (**Fig 5C**). The EPS[−] cells in this mixture were thus able to utilize EPS produced by the BPS[−] cells, indicating that EPS can be considered a "shared good" of the community. On the other hand, BPS[−] cells remained at the swarm center, suggesting that any BPS produced by the EPS[−] cells was not able to rescue the motility defect of the BPS[−] cells, suggesting that BPS may not be a "shared good."

## BPS-pathway versus EPS-pathway expression is spatially distinct within a swarm

Considering the distinct physiologies of EPS[−] versus BPS[−] swarms (**Figs 2 and 3, S2 and S3 Figs**), we set out to examine whether or not EPS and BPS were differentially regulated by cells in WT swarm communities. To first probe the spatial distribution of EPS- and BPS-pathway cluster expression within a swarm, sfGFP and mCherry were simultaneously placed under EPS- and BPS-cluster promoter control (P$_{EPS}$-sfGFP [*wzxX* promoter] and P$_{BPS}$-mCherry [*wzcB* promoter], respectively) in the same WT cell. The spatio-specific expression of the 2 reporter genes was subsequently monitored within the swarm via a fluorescence microscopy technique allowing for the acquisition of large-scale images at high resolution. Although cells plated immediately from liquid cultures ($t$ = 0) demonstrated homogeneous expression profiles of both reporters across the population, spatial separation of signals within a swarm was observed already after 24 hours (**S5 Fig**). These distinct spatial signal profiles were even more pronounced after 48 hours, with the P$_{BPS}$-mCherry signal preferentially expressed toward the swarm interior, whereas P$_{EPS}$-sfGFP signal was more highly expressed around the swarm periphery (**Fig 6A**). Fruiting bodies that appeared after 72 hours indicated signal overlap between P$_{EPS}$-sfGFP and P$_{BPS}$-mCherry expression, suggesting that EPS and BPS pathways are both active within these multicellular structures; at this time point, P$_{EPS}$-sfGFP expression was still detectable at the swarm edge (**S5 Fig**).

**Table 1.** [1]H and [13]C NMR data (δ, ppm, D$_2$O, 27˚C, 500 MHz) for the deacetylated polysaccharide from Δ*wzaX* Ω*pilA* concentrated supernatant.

| Sugar | Nucleus | H/C 1 | H/C 2 | H/C 3 | H/C 4 | H/C 5 | H/C 6 |
|---|---|---|---|---|---|---|---|
| 4-β-D-ManNAc (residue A) | [1]H | 4.90 | 4.36 | 3.90 | 3.67 | 3.47 | 3.74; 3.86 |
| | [13]C | 97.2 | 53.9 | 71.8 | 77.6 | 75.9 | 61.2 |
| 4-β-D-ManNAcA (residue B) | [1]H | 4.83 | 4.55 | 3.86 | 3.74 | 3.74 | |
| | [13]C | 100.4 | 53.8 | 71.6 | 78.9 | 77.3 | |
| 4-β-D-ManNAcA (residue C) | [1]H | 4.70 | 4.45 | 3.83 | 3.74 | 3.74 | |
| | [13]C | 99.8 | 54.0 | 71.6 | 78.9 | 77.3 | |
| 3-β-D-ManNAcA (residue D) | [1]H | 4.66 | 4.57 | 3.96 | 3.58 | 3.68 | |
| | [13]C | 99.7 | 51.1 | 78.4 | 68.6 | 77.6 | |

NAc signals (CH$_3$): 2.02/23.2; 2.02/23.2; 2.05/23.3; 2.06/23.3 ppm.

Abbreviations: D-ManNAc, *N*-acetyl-D-mannosamine; D-ManNAcA, *N*-acetyl-D-mannosaminuronic acid; ppm, parts per million.

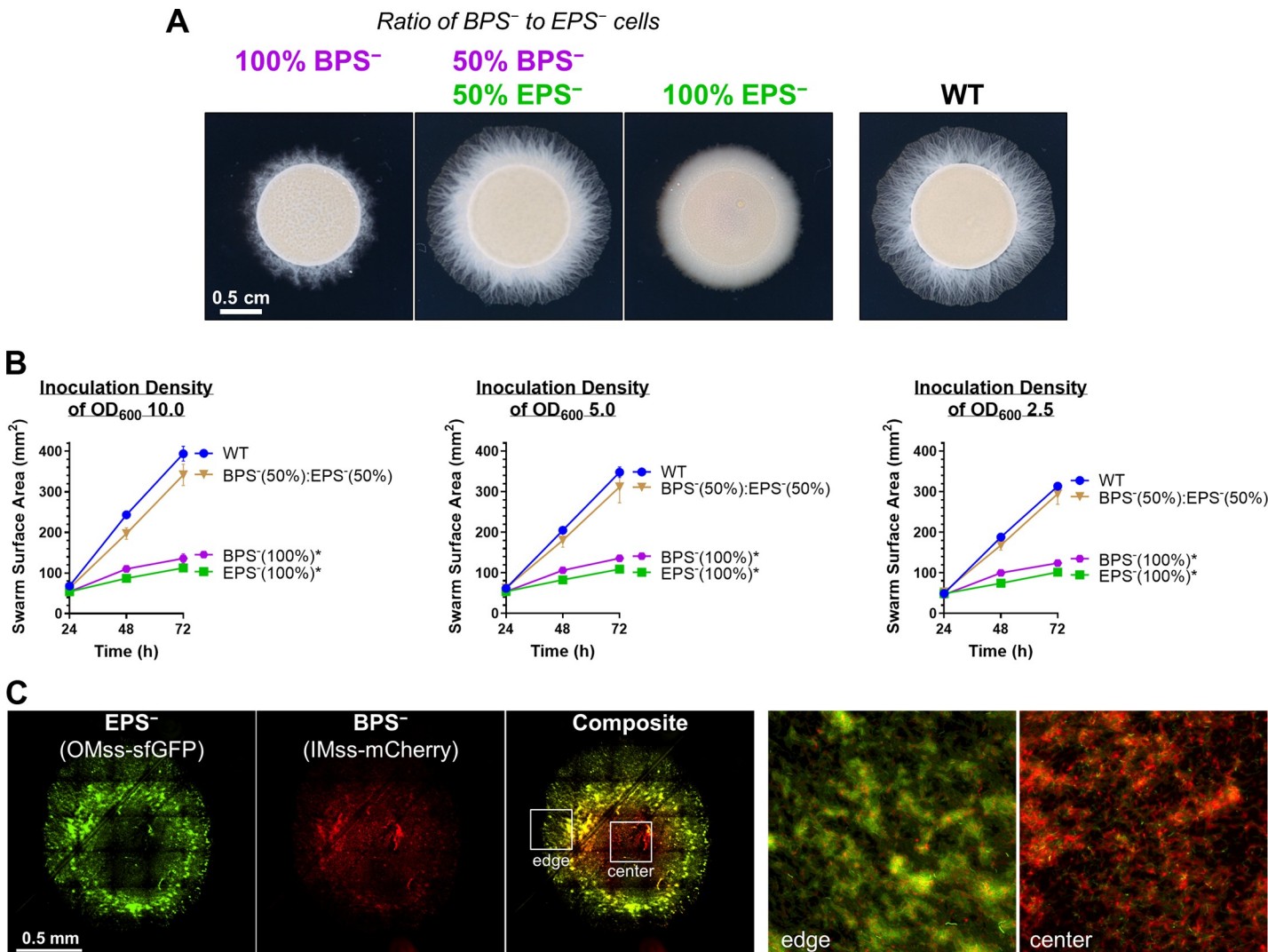

**Fig 5. Cross-complementation of EPS versus BPS deficiencies via strain mixing. (A)** EPS⁻ (Δ*wzaX*) and BPS⁻ (Δ*wzaB*) cells from exponentially growing cultures were mixed at the indicated ratios to a final concentration of OD$_{600}$ 5.0. Pure and mixed cultures were then spotted on CYE 0.5% agar and imaged after 48 hours at 32°C. **(B)** Swarm areas with temporal tracking of pure and mixed cultures were treated as described and imaged at 24, 48, and 72 hours. Each data point is the average of 4 biological replicates and is displayed +/− SEM. Mixed/pure cultures with statistically significant differences ($p \leq 0.05$) in mean surface areas (at 72 hours relative to WT) are indicated with an asterisk (*), as determined via 1-way ANOVA followed by Dunnett's multiple comparisons test. Raw values and detailed statistical analysis are available (**S1 Data**). **(C)** EPS⁻ (Δ*wzaX*) P$_{pilA}$-OMss-sfGFP and BPS⁻ (Δ*wzaB*) P$_{pilA}$-IMss-mCherry cells were mixed at a 1:1 ratio as in (A), spotted on agar pads, and imaged via fluorescence microscopy after 24 hours. The 2 images on the right are magnified views of the colony center and colony edge approximately indicated by the inset boxes in the "composite" image. BPS, biosurfactant polysaccharide; CYE, casitone-yeast extract; IMss, inner-membrane signal sequence; OMss, outer membrane signal sequence; OD$_{600}$, optical density at 600 nm; sfGFP, superfolder green fluorescent protein; WT, wild type.

To quantify the differential P$_{EPS}$-sfGFP and P$_{BPS}$-mCherry expression patterns observed via fluorescence microscopy (**Fig 6A**), cell samples were collected from the edges and the centers of swarms after 48 hours and analyzed via flow cytometry. Although red fluorescence (from P$_{BPS}$-mCherry) was 2 times more intense towards the colony interior compared with the periphery, the inverse relationship was observed for green fluorescence; P$_{EPS}$-sfGFP expression was 2 times more intense at the periphery versus the swarm center (**Fig 6B**). These flow cytometry data directly reinforce the spatial expression patterns described above (**Fig 6A**).

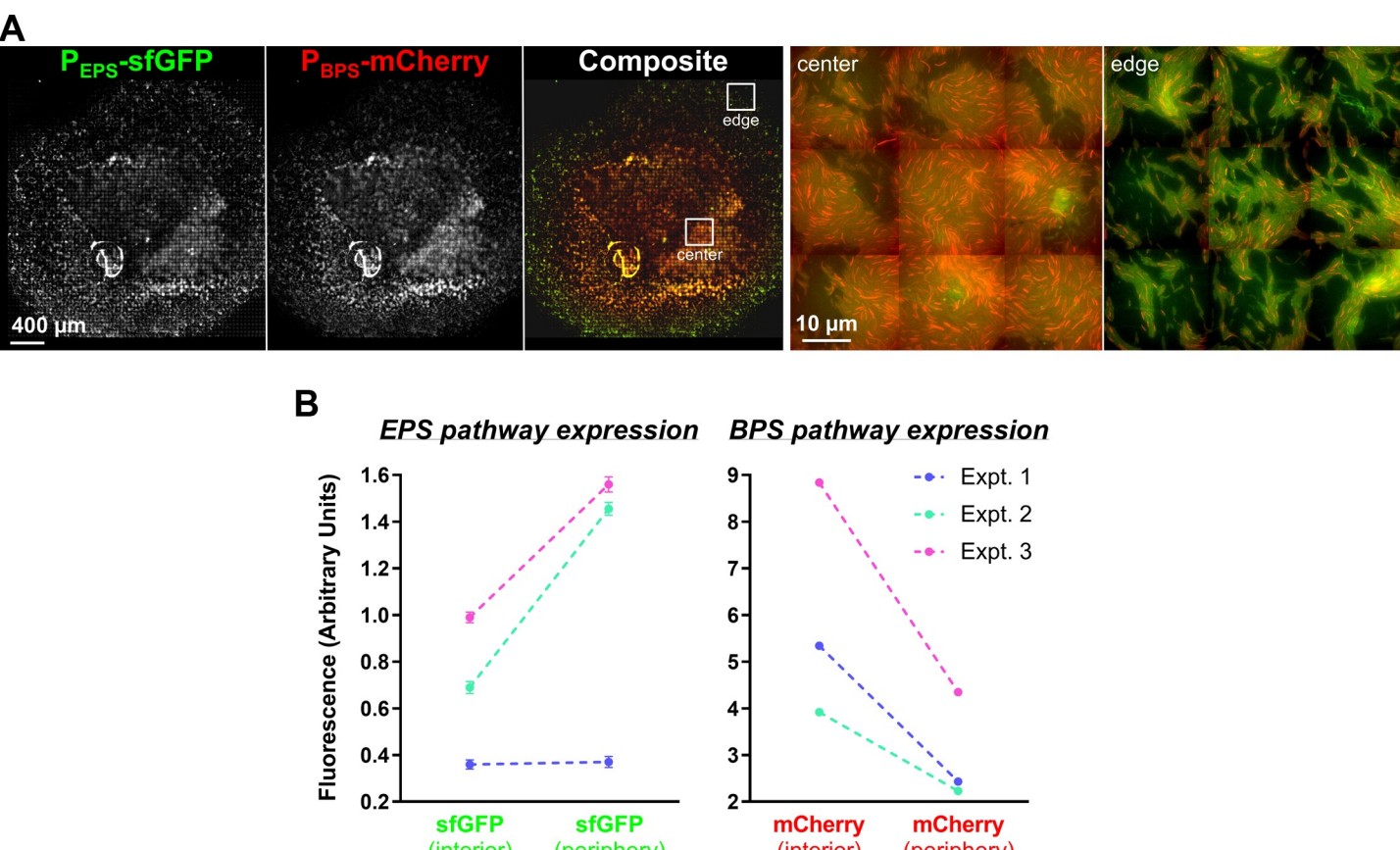

**Fig 6. Analysis of the spatial expression of the EPS and BPS gene clusters. (A)** Dual-labeled ($P_{EPS}$-sfGFP + $P_{BPS}$-mCherry) WT cells (strain EM709) from exponentially growing cultures were spotted on developmental media at a final concentration of $OD_{600}$ 5.0 and imaged at 48 hours. Images were scaled as described in "Material and methods." The 2 images on the right are magnified views of the colony center and colony edge at positions approximately indicated by the inset boxes in the composite image. **(B)** Flow cytometry analysis of WT + $P_{EPS}$-sfGFP + $P_{BPS}$-mCherry cells harvested from the colony interior or periphery following incubation for 48 hours. Cells were analyzed for intensity of sfGFP and mCherry fluorescence. Results of 3 independent experiments are displayed (first experiment, blue lines; second experiment, green lines; third experiment, magenta lines). For each experiment, a total population of 300,000–500,000 events was used and statistically analyzed. Differences between fluorescence intensity at the colony center versus edges are significant ($p < 0.0001$) for all experiments except for the first experiment with sfGFP. Errors bars are set at 1% confidence. Signals obtained with the nonfluorescent WT strain were subtracted from the fluorescence signals of strain EM709. Raw values are available (S3 Data). BPS, biosurfactant polysaccharide; Expt., experiment; EPS, exopolysaccharide; $OD_{600}$, optical density at 600 nm; sfGFP, superfolder green fluorescent protein; WT, wild type.

All together, these data indicate that the production of EPS and BPS occurs with a precise spatial regulation likely reflecting the requirement for the 2 polysaccharides in specific cell behaviors within different swarm regions. For example, EPS may be preferentially required at the lower-density colony edge where T4P-dependent swarm spreading takes place, whereas BPS is more crucial at the high-cell-density colony center to perhaps favor initial cell dispersal.

## Discussion

Given the extensive genomic, phenotypic, biochemical, and biophysical characterizations described herein, we propose that the product of the BPS assembly pathway is a novel T4P-inhibited acidic heteropolysaccharide with surface-active properties. These characteristics enhance swarm expansion. Importantly, these data directly affect past, present, and future interpretations of *M. xanthus* physiology attributed to effects on EPS, which could now be due in part to the differential regulation of BPS.

In this investigation, we have demonstrated that *M. xanthus* uses spatially distinct polysaccharide production patterns among members of the swarm community to modulate its complex multicellular lifecycle, with differently localized subpopulations of cells favoring production of EPS versus BPS to promote dissemination of the swarm community. This dynamic thus facilitates the development of complex swarm-level behaviors requiring more than the activity of a collection of single cells. It is in this manner that the collective behavior of *M. xanthus* cells in swarms leads to the differentiation of the population into forager cells, peripheral rods, and spore-forming cells.

There is an unfortunate misconception throughout much of the scientific community on the topic of biological surface-active agents [59], in which the terms "bioemulsifier" and "biosurfactant" are used interchangeably, even within the same article, despite strict definitions for what constitutes each. It bears repeating that (1) a bioemulsifier is a naturally produced compound capable of stabilizing an emulsion between 2 immiscible phases, whereas (2) a biosurfactant, in addition to being able to stabilize an emulsion, must also be able to reduce the surface/interfacial tension between 2 phases. Therefore, all biosurfactants are also by definition bioemulsifiers, but not all bioemulsifiers are biosurfactants. Within this article, we have endeavored to rigorously apply these definitions, but the overall lack of precision makes following the relevant scientific literature quite difficult.

The combined importance of secreted polymers and surface-active agents for community expansion and migration may be a common theme for bacteria, especially for species that display differentiated cell fates. In the model gram-positive bacterium *Bacillus subtilis*, at least 5 cell types have been described to date, with each distinguished via the following phenotypes: protease production, sporulation, motility, matrix production, and surfactin production [64,65]. (1) Extracellular protease production is correlated with later stages of biofilm formation, suggesting a role in nutrient acquisition or escape from a biofilm [66]. (2) Sporulation allows for *B. subtilis* to survive prolonged periods of desiccation and nutrient scarcity, whereas (3) motility phases involve synthesis of external appendages (flagella) for swimming [65]. As with numerous other bacteria, (4) *B. subtilis* also biosynthesizes an extracellular polysaccharide that constitutes the bulk of its biofilm matrix and that, under certain conditions, can affect colony expansion. (5) Finally, *B. subtilis* also famously produces surfactin, a cyclic lipopeptide with strong surfactant properties [67–69]. Recently, the formation of matrix polysaccharide-dependent "van Gogh" bundles of *B. subtilis* cells at the colony edge was shown to be greatly improved in the presence of nearby surfactin-producing cells; the net effect of this interplay between matrix polysaccharide and surfactin was thus to determine the rate of colony expansion [6]. Intriguingly, each of the differentiating phenotypes describe above has a direct parallel in *M. xanthus*. (1) Numerous proteases are released by *M. xanthus* cells and are suspected to participate in nutrient acquisition (via predation) [70], (2) whereas myxospore formation in fruiting bodies promotes survival under nutrient-limiting conditions [13]. Though single-cell gliding motility does not require any external appendages [12], (3) swarm-level motility is mediated by extension and retraction of T4P appendages; (4) this is in conjunction with the requirement of EPS for T4P-dependent swarm spreading [13]. (5) As reported herein, *M. xanthus* also produces a surface-active BPS molecule, in the form of a high-molecular-weight heteropolysaccharide polymer, with the effect of BPS on EPS modulating swarm-level structure, migration, and the overall developmental cycle of the bacterium.

The production of a secreted polysaccharide with emulsifying properties and its interaction with cell surface-associated carbohydrates also has parallels in another well-studied system. Originally isolated from a mixed culture growing on crude oil [71], *Acinetobacter* sp. RAG-1 was found to generate a product able to stably emulsify hydrocarbons [72,73] as well as reduce interfacial tension [72]. Termed RAG-1 emulsan, this compound has been extensively tested

with respect to environmental applications, including oil spill cleanup and bioremediation of heavy metal pollution [74,75]. RAG-1 emulsan had been studied for over 35 years [71] before a more rigorous extraction protocol was developed, revealing that RAG-1 emulsan was in fact a bipartite compound composed of (1) a rough-type LPS and (2) a high-molecular-weight secreted polysaccharide [76]. The latter compound, now termed *Acinetobacter* polyelectrolytic exopolysaccharide (APE) [77], is synthesized via a Wzx/Wzy-dependent pathway [78,79]; it is the component responsible for the emulsifying properties of RAG-1 emulsan [76], but to our knowledge, the capacity of purified APE to reduce interfacial tension has never been reported. Thus, the role of APE as a true biosurfactant has yet to be established. As the APE polymer is built from repeating →4)-D-GalNAc-6-OAc-(α1→4)-D-GalNAcA-(α1→3)-D-QuiNAc4NHb-(α1→ trisaccharide units [77], it is conceivable that the presence of uronic acid and acetyl moieties are able to contribute hydrophilic and hydrophobic character to APE, respectively. Because *M. xanthus* BPS is composed of →3)-D-ManNAcA-(β1→4)-D-ManNAcA-(β1→4)-D-ManNAcA-(β1→4)-D-ManNAc-(β1→ repeating units, considerable hydrophilic and hydrophobic character would again be present due to its high uronic acid content as well as its *N*-/*O*-acetylation levels, respectively. Together, these traits could explain the emulsifying properties of both APE and BPS (as well as the demonstrated surface tension-reducing surfactant properties in the latter). Similarly to *Acinetobacter* and RAG-1 emulsan, physiology connected directly to the presence/absence of "EPS" in *M. xanthus* has been studied for over 30 years [80]; however, the complex physiology of this social bacterium must now be considered within the context of the interplay between the dedicated EPS polymer on the cell surface and the newly identified secreted BPS product.

With the identification of BPS and its importance to *M. xanthus* physiology, numerous questions have been raised. For instance, given the spatio-specific differences in production of the various polymers, by what mechanism is BPS production regulated in relation to EPS and MASC? The presence of functionally important BYK homologues to the production of BPS/EPS/MASC represents an enticing level of production control for each of the polymers. In addition, the apparent suppression of BPS production by the presence of a T4P suggests possible links between BPS and the Dif pathway, with the latter already known to regulate T4P-dependent EPS expression. As biosurfactants are of immense industrial interest and importance, further characterization of the chemical properties of BPS are also immediate avenues of future inquiry.

Ultimately, our investigation reveals that differentiated functions between distinct cell subpopulations across an entire swarm generates an ecologically beneficial, community-level, higher-order organization. This is a central tenet governing the varied evolutionary origins of multicellularity in nature.

## Note added in proof

During the submission-and-review process for this manuscript, Pérez-Burgos and colleagues independently annotated the *M. xanthus* MASC-pathway WzxS flippase and WzyS polymerase identified herein and illustrated the functional requirement of these proteins in sporulation [81], independently supporting our findings for the MASC pathway.

## Materials and methods

### Bacterial cell culture

The *M. xanthus* strains used in this study are listed in **S3 Table**. They were grown and maintained at 32˚C on casitone-yeast extract (CYE) agar plates or in CYE liquid medium at 32˚C on a rotary shaker at 160 rpm. The *E. coli* strains used for plasmid construction were grown

and maintained at 37˚C on LB agar plates or in LB liquid medium. Plates contained 1.5% agar (BD Difco). Kanamycin at 100 μg/mL and galactose at 2.5% (w/v) were added to media for selection when appropriate.

## Plasmid and mutant construction

Plasmids used in this study are listed in **S3 Table**. To create *M. xanthus* in-frame deletion strains, 900 bp upstream and downstream of the gene targeted for deletion were amplified and fused via PCR, restriction digested, then ligated into pBJ113 or pBJ114 [82]. The resulting plasmids were then introduced into WT *M. xanthus* DZ2 via electroporation. Mutants resulting from homologous recombination of the deletion alleles were obtained by selection on CYE agar plates first containing kanamycin and then containing galactose to resolve the merodiploids.

To fluorescently label the IM or OM of the *M. xanthus* EPS⁻ (Δ*wzaX*) and BPS⁻ (Δ*wzaB*) strains, cells were electroporated with the integrative vector pSWU19 encoding either the OMss-sfGFP or the IMss-mCherry fusion, respectively, under control of the *pilA* promoter [18]. Strain EM709 was obtained by amplifying 1,000 bp upstream of the start codon of genes *mxan_7416* and *mxan_1025* (https://www.genome.jp/kegg/) and fusing them to the *sfgfp* or *mcherry* coding sequences, respectively. The 2 gene fusions were then cloned into the same pSWU19 vector. The recombinant vector was then transferred via electroporation into *M. xanthus* DZ2, thus inserting the 2 gene fusions together at the Mx8 phage-attachment site in the chromosome [18] and allowing for co-expression in *M. xanthus*.

## Phylogeny and gene co-occurrence

Forty order Myxococcales genomes [83–98] were downloaded from NCBI followed by RAST-based gene prediction and annotation [99]. Thirty housekeeping proteins (DnaG, Frr, InfC, NusA, Pgk, PyrG, RplC, RplD, RplE, RplF, RplK, RplL, RplM, RplN, RplP, RplS, RplT, RpmA, RpoB, RpsB, RpsC, RpsE, RpsI, RpsJ, RpsK, RpsM, RpsS, SmpB, Tsf) were aligned, concatenated, and subjected to FastTree 2.1.8 (http://www.microbesonline.org/fasttree/) analysis to generate a maximum-likelihood phylogeny with 100 bootstrap values using the Jones–Taylor–Thornton (JTT) protein substitution model. Functional domains were identified by scanning all proteomes against the Pfam-A v29.0 database [100] (downloaded: Oct 26, 2016) using hmmscan (E-value cutoff 1e-5) from the HMMER suite (http://hmmer.janelia.org/) [101] and further parsed using hmmscan-parser.sh. Pfam domains attributed to Wzx (MXAN_7416; PF13440 [Polysacc_synt]), Wzy (MXAN_7442; PF04932 [Wzy]), Wzc (MXAN_7421; PF02706 [Wzz]), and Wza (MXAN_7417; PF02563 [Poly_export]) were identified, and the protein information was extracted. Based on the identified proteins and the location in the genome of their coding sequences, clusters were manually curated. Along with Pfam domain analysis, we used all protein sequences forming identified clusters to perform protein Basic Local Alignment Search Tool (BLASTp) searches [102] against the predicted proteome of each organism; as sequence identity and similarity are quite low between homologues of Wzx/Wzy-dependent pathway proteins (even within the same species) [29], looser-stringency cutoffs were used for these searches (E-value of 0.00001, 35% query coverage and 35% similarity). To confirm the participation of identified homologues (via Pfam and BLAST) in respective clusters, maximum-likelihood phylogenetic trees (JTT Model and 100 bootstrap values using FastTree 2.1.8) were generated of homologues to MXAN_7416, MXAN_7417, MXAN_7421, and MXAN_7442. Finally, the binary distribution (presence/absence), location, and cluster information of each cluster were mapped on the housekeeping protein-based phylogeny. To support protein annotations, consensus α-helical TMS prediction [103] was

obtained via OCTOPUS [104] and TMHMM [105], followed by fold-recognition analyses performed using HHpred [106].

## Phenotypic analysis

Exponentially growing cells were harvested and resuspended in TPM buffer (10 mM Tris-HCl [pH 7.6], 8 mM $MgSO_4$, and 1 mM $KH_2PO_4$) at a final concentration of $OD_{600}$ 5.0 for swarming assays or $OD_{600}$ 1.5, 2.5, 5.0, and 10.0 for developmental assays. This cell suspension (10 μL) was spotted onto CYE 0.5% agar or CF 1.5% agar for swarming or developmental (i.e., fruiting body formation) assays, respectively. Swarming plates were incubated at 32˚C for 48 hours and photographed with an Olympus SZ61 binocular stereoscope. Fruiting body plates were incubated at 32 ºC for 72 hours and photographed with an Olympus SZX16 binocular stereoscope. Contours of swarms for T4P-dependent motility quantitation (**Fig 2A**) were defined in FIJI and analyzed for total surface area.

To examine the effect of exogenous biosurfactant on swarming, 100 μL of *B. thailandensis* E264 di-rhamnolipid-$C_{14}$-$C_{14}$ stock solution (300 ppm, 0.2 $μm^2$-filtered) was first spread on top of a square (15 cm × 15 cm) CYE 0.5% agar surface and allowed to dry in a biohood prior to inoculation. To amplify spreading differences between strains complemented or not by rhamnolipid addition, 5 μL of $OD_{600}$ 5.0 TPM resuspensions were spotted, followed by incubation at 32˚C for 72 hours. For strain-mixing time-course experiments, 10 μL of TPM resuspensions at $OD_{600}$ 2.5, 5.0, and 10.0 were spotted on square CYE 0.5% agar plates, with all 4 samples for each particular $OD_{600}$ spotted on the same plate. Four biological replicates were analyzed for each set of parameters. Plates were inverted and incubated at 32˚C, with swarm images captured at 24-, 48-, and 72-hour time points. For rhamnolipid-addition and swarm-mixing assays, images were captured using an Olympus SZX16 stereoscope with UC90 4K camera; swarm contours in each image were then defined using cellSens software (Olympus), followed by calculation of surface area.

## Fluorescence microscopy

For fluorescence microscopy imaging of swarms, exponentially growing cells were first harvested and resuspended as pure or mixed cultures in TPM buffer at a final concentration of $OD_{600}$ 5.0. This cell suspension (1 μL) was spotted onto thin pads of CYE 1% agar for swarming assays and CF 1.5% agar for developmental assays. To avoid the desiccation of the thin agar pads, the agar was poured onto squared adhesive frames previously pasted on to glass slides. Slides were then incubated at 32˚C for 0, 24, 48, and 72 hours, covered with the coverslips, and photographed with a Nikon Eclipse TE2000 E PFS inverted epifluorescence microscope. Slides were imaged with a 10× objective for the strain-mixing experiments, and with a 100× objective for imaging of the dual-labeled strain. For each slide, a series of images was automatically captured by the aid of the Nikon Imaging Software to cover a section of the swarm via tiling and stitching. The microscope devices were optimized in order to minimize the mechanical movement and provide rapid autofocus capability (epi/diascopic diode lightening, piezoelectric stage) as previously described [82]. The microscope and devices were driven by the Nikon-NIS "JOBS" software [82]. For analysis of cells expressing mCherry and sfGFP transcriptional fusions (**Fig 6A**), images were scaled as follows: from the raw images, the background fluorescence intensity and mean fluorescence intensity were subtracted in order to scale the 2 images to the same mean value. The fluorescence intensity of each was then divided by its standard deviation to also scale the intensities of the fluctuations. The described processing was performed with FIJI software. The raw, untreated images are available (**S5B Fig**).

### Trypan blue dye retention

Trypan blue dye-retention analysis was adapted from a previous report [57]. Cells from overnight cultures were sedimented and resuspended in TPM buffer to $OD_{600}$ 1.0, after which 900 μL of cell resuspension was transferred to a 1.5-mL microfuge tube; a cell-free blank was also prepared with an identical volume of TPM. To each tube, 100 μL of trypan blue stock solution (100 μg/mL) was added, followed by a brief 1-second pulse via vortex to mix the samples. All tubes were placed in a rack, covered with aluminum foil, and incubated at room temperature on a rocker platform for 1 hour. After this dye-binding period, samples were sedimented at high speed in a tabletop microfuge (16,000$g$, 5 minutes) to clear supernatants of intact cells. So as not to disrupt the pellet, only 900 μL of the dye-containing supernatant was aspirated and transferred to a disposable cuvette. The spectrophotometer was directly blanked at 585 nm (the absorption peak for trypan blue) using the cell-free "TPM + trypan blue" sample to calibrate the baseline absorbance corresponding to no retention of trypan blue dye by cells. Absorbance values at 585 nm ($A_{585}$) were obtained for each clarified supernatant and normalized as a percentage of the WT $A_{585}$ reading (1) as an internal control for each individual experiment and (2) to facilitate comparison of datasets across multiple biological replicates. Negative final values are due to trace amounts of cell debris detected at 585 nm in individual samples in which absolutely no binding of trypan blue occurred.

### Purification and monosaccharide analysis of cell-associated sugars

*M. xanthus* cell-associated sugars were purified from CYE 0.5% agar-grown cultures as described [107], with the following modifications. Cell cultures were harvested and resuspended in 25 mL TNE buffer (100 mM Tris [pH 7.5], 100 mM NaCl, 5 mM EDTA). Cells in suspension were then disrupted via sonication (4 pulses, 15 seconds each), followed by addition of SDS to a final concentration of 0.1% to extract cell-associated sugars. To remove DNA from cell samples, lysates were treated with 20 μL of 10 kU DNase I from LG Healthcare (20 mM Tris-HCl [pH 7.6], 1 mM $MgCl_2$, 50% v/v glycerol) and incubated at room temperature for 5 minutes. To obtain protein-free extracellular sugar samples, 1 mg/mL of Pronase E protease mixture (Sigma-Aldrich) was gently added directly and allowed to incubate at 37˚C (2 hours). The extracts were sedimented (10 minutes, 7,513$g$, 17˚C), and the pellets were washed twice with 25 mL TNE+SDS. The pellets were then washed twice with 10 mL TNE to remove any remaining SDS. To remove EDTA, sugar samples were washed twice with MOPS buffer (10 mM MOPS [pH 7.6], 2 mM $MgSO_4$) and twice with cohesion buffer (10 mM MOPS buffer [pH 6.8], 1 mM $CaCl_2$ 1 mM $MgCl_2$). Finally, sugar samples were stored in cohesion buffer at −80˚C until use.

Cell-associated sugars purified from colonies (50 μL) were mixed with 500 μL of 12 M $H_2SO_4$ and incubated for 1 hour at 37˚C under mild shaking; 20 μL of each sample were then mixed with 220 μL of distilled water, and the diluted samples were autoclaved for 1 hour at 120˚C. After cooling, 50 μL of 10 M NaOH was added, and the samples were sedimented (10,000$g$, 10 minutes, room temperature) [108]. Supernatant (5 μL) was mixed with $ddH_2O$ (245 μL), followed by identification and quantification of the released monosaccharides via high-performance anion-exchange chromatography coupled with pulsed amperometric detection (HPAEC-PAD), performed in a Dionex ICS 3000 (Thermo Scientific) equipped with a pulsed amperometric detector. Sugar standards or EPS hydrolysates (25 μL) were applied to a Dionex CarboPac PA20 column (3 × 150 mm) preceded by the corresponding guard column (3 × 30 mm) at 35˚C. Sugars were eluted at 0.45 mL/min with the buffers 0.1 M NaOH, 1 M sodium acetate + 0.1 M NaOH and $ddH_2O$ as the eluents A, B, and C, respectively. The following multistep procedure was used: isochratic separation (10 minutes, 27% A + 73% C),

gradient separation (20 minutes, 2–19% B + 98–81% C), column wash (5 minutes, 100% A), and subsequent column equilibration (10 minutes, 27% A + 73% C). Injection of samples containing glucose, rhamnose, *N*-acetylglucosamine, arabinose, xylose, mannose, galactose, and glucosamine (Sigma-Aldrich) at known concentrations (ranging from 5 to 100 μM) was used to identify and quantify the released monosaccharides.

## Purification and analysis of secreted polysaccharides

Lyophilized concentrated samples representing 175 mL of original culture supernatant for each strain were resuspended in ddH$_2$O and treated with 2% acetic acid (10 minutes, 80˚C) to precipitate proteins and nucleic acids. Solutions were then separated via gel chromatography on a Sephadex G-15 column (1.5 cm × 60 cm) or Biogel P6 column (2.5 cm × 60 cm), in 1% acetic acid, monitored by refractive index detector (Gilson).

Anion-exchange chromatography was then performed via sample injection into a HiTrapQ column (Amersham, 2 columns × 5 mL each, connected together) in ddH$_2$O at 3 mL/minute. Samples were washed with ddH$_2$O for 5 minutes, then eluted with a linear gradient from ddH$_2$O to 1 M NaCl over 1 hour with UV detection at 220 nm. Spot tests were performed on silica TLC plates, developed by dipping in 5% H$_2$SO$_4$ in ethanol and heating with heat gun until brown spots became visible. Samples were desalted on a Sephadex G-15 column.

NMR experiments were carried out on a Varian INOVA 500 MHz ($^1$H) spectrometer with 3-mm Z-gradient probe with acetone internal reference (2.225 ppm for $^1$H and 31.45 ppm for $^{13}$C) using standard pulse sequences for gCOSY (gradient selected correlation spectroscopy), TOCSY (mixing time 120 milliseconds), ROESY (rotating frame nuclear Overhauser effect spectroscopy) (mixing time 500 milliseconds), and gHSQCAD (gradient heteronuclear single quantum correlation adiabatic). Resolution was kept <3 Hz/pt in F2 in proton–proton correlations and <5 Hz/pt in F2 of H–C correlations. The spectra were processed and analyzed using the Bruker Topspin 2.1 program.

Monosaccharides were identified by COSY, TOCSY, and NOESY cross-peak patterns and $^{13}$C NMR chemical shifts. Aminogroup location was concluded from the high-field signal position of aminated carbons (CH at 45–60 ppm).

Electrospray ionization (ESI) mass spectrometry (MS) was performed using a Waters SQ Detector 2 instrument. Samples were injected in 50% MeCN with 0.1% TFA.

## Emulsification testing

Overnight *M. xanthus* cultures (50 mL CYE in 250 mL flasks) were inoculated at an initial OD$_{600}$ of 0.05 and grown at 32˚C with shaking (220 rpm) to saturation (OD$_{600}$, approximately 5.0–7.0). Cultures were transferred to a 50-mL conical tube and sedimented at 7,000*g* (25 minutes, 22˚C, JA-17 rotor). Supernatants were decanted into a syringe, passed through a 0.22-μm filter to remove remaining cells, and transferred (4 mL) to a quartz cuvette, followed by addition of 300 μL hexadecane (Sigma) colored with Sudan Black dye (0.1 g of Sudan Black powder per 50 mL of hexadecane). Each cell-free supernatant sample was vigorously mixed with the colored hexadecane 250 times over 2 minutes (via aspiration/ejection with a p1000 micropipette). Cuvettes were then immediately inserted into a spectrophotometer, with continual, rapid manual attempts made to obtain an initial OD$_{600}$ reading, with this time recorded. After obtaining an initial OD$_{600}$ reading, subsequent OD$_{600}$ readings were manually carried out at 20-second intervals over 10 minutes to monitor the rate of emulsion clearance. All OD$_{600}$ readings for each time course were normalized with respect to the initial OD$_{600}$ value detected for each sample.

## Surface tension testing

The adsorption and interfacial properties as a function of time for supernatants of the 5 strains (and their secreted polysaccharides) were analyzed by means of a digital Tracker Drop Tensiometer (Teclis, Civrieux-d'Azergues, France) [109] at room temperature. From digital analysis of a liquid drop or an air bubble profile collected by a high-speed CCD camera, characteristic parameters (surface tension, area, volume) were determined in real time. Surface tension was estimated from the Laplace equation adapted for a bubble/drop. By controlled movements of the syringe piston, driven by a step-by-step motor, surface area can be maintained constant during the whole experiment. Before each experiment, cleanliness of material was tested using ultrapure water, before being dried with argon. For the study of tensioactive properties of the supernatants, a 10-μL air bubble was formed at the tip of a J-tube submerged in 5 mL of supernatants for each strain.

## Flow cytometry

The fluorescence intensities of *M. xanthus* strain EM709 simultaneously expressing $P_{EPS}$-sfGFP and $P_{BPS}$-mCherry were measured by flow cytometry with a Bio-Rad S3E cells sorter. The blue laser (488 nm, 100 mW) was used for the forward scatter (FSC), side scatter (SSC), and excitation of sfGFP, whereas the green laser (561 nm, 100 mW) was used for the excitation of mCherry. Signals were collected using the emission filters FL1 (525/30 nm) and FL3 (615/25 nm) for sfGFP and mCherry, respectively. Cells collected from the colony edges and centers were suspended in TPM and ran at low-pressure mode and at a rate of 10,000 particles per second. The threshold on FSC was 0.12, and the voltages of the photomultipliers were 361, 280, 785, and 862 volts for FSC, SSC, FL1, and FL3, respectively. The density plots obtained (small-angle scattering FSC versus wide angle scattering SSC signals) were gated on the population of interest and filtered to remove multiple events. Populations of 300,000 to 500,000 events were used and analyzed statistically using the FlowJo software. The sfGFP and mCherry signals obtained with the nonfluorescent WT cells were subtracted from the signals obtained with cells of strain EM709. Measurements were carried out 3 times with bacteria from different plates.

## Supporting information

**S1 Fig. Gene conservation and synteny diagrams for *M. xanthus* Wzx/Wzy-dependent pathway clusters. (A)** EPS-, **(B)** MASC-, and **(C)** BPS-cluster evolution: core assembly-and-export constituents of each biosynthesis pathway were identified using Pfam domain scanning and BLAST-based homology searches. Based on the genomic locations of core and other neighborhood genes in *M. xanthus*, clusters were generated, and their binary distribution was further mapped on to the phylogenetic tree generated via aligning and concatenating 30 housekeeping proteins as shown in this figure. Arrows represent the uninterrupted presence of genes in a cluster. Locus tags highlighted by pale blue boxes correspond to genes such as enzymes involved in monosaccharide synthesis, modification, or incorporation into precursor repeat units of the respective polymer. White circles depict the presence of a homologous gene encoded elsewhere in the chromosome (but not syntenic with the remainder of the EPS/MASC/BPS biosynthesis cluster). Bootstrap values are provided on the tree nodes. BLAST, Basic Local Alignment Search Tool; BPS, biosurfactant polysaccharide; EPS, exopolysaccharide; MASC, major spore coat polysaccharide.
(PDF)

**S2 Fig. Effects of EPS and BPS deficiencies on *M. xanthus* predatory and density-dependent developmental outcomes.** **(A)** Fruiting body formation phenotypes at different cell densities after growth at 32˚C for 72 hours. **(B)** WT, EPS⁻ (Δ*wzaX*), and BPS⁻ (Δ*wzaB*) cells from exponentially growing cultures were resuspended in buffer to a final concentration of $OD_{600}$ 10. Samples were then spotted on developmental media next to an *E. coli* colony and imaged after 48 hours. The first row contains images of the entire swarm; the second row presents a medium-magnification view of the prey invasion step; the third row presents a high-magnification view of the ripples at sites corresponding to the small boxes on the second row. BPS, biosurfactant polysaccharide; EPS, exopolysaccharide; $OD_{600}$, optical density at 600 nm; WT, wild type.
(PDF)

**S3 Fig. Analyses of cell-associated versus secreted characteristics related to BPS deficiency.** **(A)** High-performance anion-exchange chromatography coupled with pulsed amperometric detection for monosaccharide standards versus monosaccharides isolated from submerged-culture supernatants. Strains tested: WT and BPS⁻ (Δ*wzaB*). **(B)** Boxplots of trypan blue dye retention to indicate the levels of EPS production in various *pilA* mutant strains relative to WT. The lower and upper boundaries of the boxes correspond to the 25th and 75th percentiles, respectively. The median (line through center of boxplot) and mean (+) of each dataset are indicated. Lower and upper whiskers represent the 10th and 90th percentiles, respectively; data points above and below the whiskers are drawn as individual points. Asterisks denote datasets displaying statistically significant differences in distributions ($p < 0.05$) shifted higher (*) than WT, as determined via Wilcoxon signed-rank test performed relative to "100" (i.e., WT); stars denote datasets that are not statistically different relative to "0" ($p > 0.05$), as determined via Wilcoxon signed-rank test performed relative to "0." Raw values and detailed statistical analysis are available (**S2 Data**). **(C)** Time course of raw surface tension values (via digital-drop tensiometry) from representative submerged-culture supernatants. Strains tested: WT, MASC⁻ (Δ*wzaS*), BPS⁻ MASC⁻ (Δ*wzaB* Δ*wzaS*), EPS⁻ MASC⁻ (Δ*wzaX* Δ*wzaS*), EPS⁻ BPS⁻ MASC⁻ (Δ*wzaX* Δ*wzaB* Δ*wzaS*). **(D)** Representative images of T4P-dependent motility for WT, EPS⁻ (Δ*wzaX*), BPS⁻ (Δ*wzaB*), and MASC⁻ (Δ*wzaS*) strains in the presence/absence of di-rhamnolipid-$C_{14}$-$C_{14}$ produced by *B. thailandensis* E264 (scale bar: 4 mm). Top: Samples spotted (5 µL of $OD_{600}$ 5.0 resuspension in TPM buffer) on a CYE 0.5% agar plate, grown for 72 hours in the absence of rhamnolipid. *Bottom*: Samples treated analogously to those in top portion of the panel but grown for 72 hours on a CYE 0.5% agar plate pretreated with rhamnolipid. Analysis of these data (**Fig 3D**) as well as raw values and detailed statistical analysis (**S1 Data**) are available. BPS, biosurfactant polysaccharide; CYE, casitone-yeast extract; EPS, exopolysaccharide; MASC, major spore coat; $OD_{600}$, optical density at 600 nm; T4P, type IV pilus; TPM, Tris-phosphate-magnesium; WT, wild type.
(PDF)

**S4 Fig. Isolation and NMR analysis of natively acetylated PS isolated from Δ*wzaX* Ω*pilA* supernatant.** **(A)** Gel chromatography separation of enriched supernatant from a Δ*wzaX* Ω*pilA* culture. **(B)** $^1$H–$^{13}$C HSQC NMR spectrum of acidic PS isolated from Δ*wzaX* Ω*pilA* supernatant. Analysis was performed at 25 ºC, 500 MHz. Resonance peak colors: black, C–H; green, C–$H_2$. HSQC, heteronuclear single quantum correlation; PS, polysaccharide.
(PDF)

**S5 Fig. Time-course analysis of the spatial expression of the EPS and BPS gene clusters.** **(A)** Dual-labeled ($P_{EPS}$-sfGFP + $P_{BPS}$-mCherry) WT cells (strain EM709) from exponentially growing cultures were spotted on developmental media at a final concentration of $OD_{600}$ 10.0

and imaged at the indicated time points (scale bar: 100 μm). Images were scaled as described in "Material and methods." **(B)** Raw, nonnormalized data displayed in Panel A. BPS, biosurfactant polysaccharide; EPS, exopolysaccharide; $OD_{600}$, optical density at 600 nm; sfGFP, superfolder green fluorescent protein; WT, wild type.
(PDF)

**S1 Table. Gene nomenclature and annotation analyses for *M. xanthus* EPS, MASC, and BPS Wzx/Wzy-dependent pathway proteins.** BPS, biosurfactant polysaccharide; EPS, exopolysaccharide; MASC, major spore coat.
(PDF)

**S2 Table. Function prediction via fold-recognition analysis of EPS, MASC, and BPS synthesis cluster proteins.** Using HHpred, protein sequences were searched against entries in the Protein Data Bank, SCOP database, COG database, and Pfam database. Color schemes match those in **Fig 1B** and **S1 Fig**. BPS, biosurfactant polysaccharide; COG, clusters of orthologous groups; EPS, exopolysaccharide; MASC, major spore coat; SCOP, structural classification of proteins.
(XLSX)

**S3 Table. Bacterial strains and plasmids used in this study.**
(PDF)

**S1 Data. Source values and statistical analyses for datasets reporting swarm surface areas.** This spreadsheet covers the source values for **Fig 3A**, **Fig 3B**, **Fig 3E**, and **S3B Fig**.
(XLSX)

**S2 Data. Source values and statistical analyses for datasets involving BPS analysis.** This spreadsheet covers the source values for **Fig 2A**, **Fig 3D**, and **Fig 5B**. BPS, biosurfactant polysaccharide.
(XLSX)

**S3 Data. Source values for analyses involving a digital-drop tensiometer or a flow cytometer.** This spreadsheet covers the source values for **Fig 3C**, **Fig 6B**, and **S3C Fig**.
(XLSX)

## Acknowledgments

The authors would like to thank several individuals: (1) Mariamichela Lanzilli for constructing strain EM709; (2) Lucie Lacombe for acquisition optimization of large-scale fluorescence microscopy images; (3) Éric Déziel for insightful discussions, troubleshooting regarding emulsifier testing, providing rhamnolipid, as well as critical reading of the manuscript; (4) Alec McDermott for assistance with dye-binding assays; and (5) Philippe Constant for valuable input on biostatistics.

## Author Contributions

**Conceptualization:** Salim T. Islam, Leon Espinosa, Gael Brasseur, Jean-François Guillemot, Anaïs Benarouche, Jean-Luc Bridot, Henri-Pierre Fierobe, Tâm Mignot, Emilia M. F. Mauriello.

**Data curation:** Leon Espinosa, Gael Brasseur.

**Formal analysis:** Salim T. Islam, Israel Vergara Alvarez, Fares Saïdi, Annick Guiseppi, Evgeny Vinogradov, Gaurav Sharma, Leon Espinosa, Castrese Morrone, Gael Brasseur, Jean-

François Guillemot, Anaïs Benarouche, Jean-Luc Bridot, Henri-Pierre Fierobe, Emilia M. F. Mauriello.

**Funding acquisition:** Salim T. Islam, Evgeny Vinogradov, Alain Cagna, Mitchell Singer, Tâm Mignot, Emilia M. F. Mauriello.

**Investigation:** Salim T. Islam, Israel Vergara Alvarez, Fares Saïdi, Annick Guiseppi, Evgeny Vinogradov, Gaurav Sharma, Castrese Morrone, Jean-Luc Bridot, Gokulakrishnan Ravicoularamin, Henri-Pierre Fierobe, Emilia M. F. Mauriello.

**Methodology:** Salim T. Islam, Israel Vergara Alvarez, Fares Saïdi, Evgeny Vinogradov, Gaurav Sharma, Leon Espinosa, Gael Brasseur, Jean-François Guillemot, Jean-Luc Bridot, Henri-Pierre Fierobe, Emilia M. F. Mauriello.

**Project administration:** Salim T. Islam, Tâm Mignot, Emilia M. F. Mauriello.

**Resources:** Salim T. Islam, Evgeny Vinogradov, Leon Espinosa, Gael Brasseur, Jean-Luc Bridot, Charles Gauthier, Henri-Pierre Fierobe, Tâm Mignot, Emilia M. F. Mauriello.

**Software:** Gaurav Sharma, Leon Espinosa, Jean-François Guillemot.

**Supervision:** Salim T. Islam, Leon Espinosa, Jean-Luc Bridot, Charles Gauthier, Mitchell Singer, Tâm Mignot, Emilia M. F. Mauriello.

**Validation:** Salim T. Islam, Israel Vergara Alvarez, Fares Saïdi, Annick Guiseppi, Evgeny Vinogradov, Gaurav Sharma, Leon Espinosa, Gael Brasseur, Anaïs Benarouche, Jean-Luc Bridot, Gokulakrishnan Ravicoularamin, Charles Gauthier, Henri-Pierre Fierobe, Tâm Mignot, Emilia M. F. Mauriello.

**Visualization:** Salim T. Islam, Evgeny Vinogradov, Gaurav Sharma, Anaïs Benarouche, Henri-Pierre Fierobe, Tâm Mignot, Emilia M. F. Mauriello.

**Writing – original draft:** Salim T. Islam, Evgeny Vinogradov, Emilia M. F. Mauriello.

**Writing – review & editing:** Salim T. Islam, Henri-Pierre Fierobe, Tâm Mignot, Emilia M. F. Mauriello.

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
