## [Editor Report · Decision Letter 0]

25 Feb 2020

Dear Dr Islam, 

Thank you for submitting your manuscript entitled "Modulation of bacterial multicellularity via spatiotemporal polysaccharide secretion" for consideration as a Research Article by PLOS Biology.

Your manuscript has now been evaluated by the PLOS Biology editorial staff, as well as by an academic editor with relevant expertise, and I'm writing to let you know that we would like to send your submission out for external peer review.

IMPORTANT: We note that you have excluded six significant experts in the field as reviewers. I'm afraid that we do not feel that this is reasonable, and may not be able to honour these. It may help if you could reduce the number of exclusions or rank them in order of preference.

Please re-submit your manuscript within two working days, i.e. by Feb 27 2020 11:59PM.

Kind regards,

Roli Roberts

Senior Editor

PLOS Biology

---

## [Decision Letter · Decision Letter 1]

8 Apr 2020

Dear Dr Islam,

Thank you very much for submitting your manuscript "Modulation of bacterial multicellularity via spatiotemporal polysaccharide secretion" for consideration as a Research Article by PLOS Biology. As with all papers reviewed by the journal, yours was evaluated by the PLOS Biology editors as well as by an Academic Editor with relevant expertise and in this case by three independent reviewers.

Based on the reviews, we will probably accept this manuscript for publication, assuming that you will modify the manuscript to address the remaining points raised by the reviewers.

IMPORTANT:

a) The Academic Editor asked me to emphasise the following "Some of the claims of first are apparently exaggerated (two reviewers point this out). This certainly needs to be toned down or clarified."

b) Please also make sure to address the Data Policy requests noted further down the email.

We expect to receive your revised manuscript within two weeks. Your revisions should address the specific points made by each reviewer. In addition to the remaining revisions and before we will be able to formally accept your manuscript and consider it "in press", we also need to ensure that your article conforms to our guidelines. A member of our team will be in touch shortly with a set of requests. As we can't proceed until these requirements are met, your swift response will help prevent delays to publication.

*Copyediting*

*Published Peer Review History*

*Early Version*

*Submitting Your Revision*

Sincerely,

Roli Roberts

Senior Editor

PLOS Biology

DATA POLICY:

Regardless of the method selected, please ensure that you provide the individual numerical values that underlie the summary data displayed in the following figure panels as they are essential for readers to assess your analysis and to reproduce it: Figs 2A, 3ABCDE, 5B, 6B, and S3B. I note that data for 2A, 3D, 5B are available in Supp Table 4, and that data for 3A and S3B are in Supp Table 5. These two files should be renamed Supp Data 1 and Supp Data 2 and cited in the relevant Fig legends. I’m not seeing data for Figs 3BCE or 6B; please supply these or clrify where they can be found. NOTE: the numerical data provided should include all replicates AND the way in which the plotted mean and errors were derived (it should not present only the mean/average values).

REVIEWERS' COMMENTS:

Reviewer #1:

The manuscript entitled « Modulation of bacterial multicellularity via spatiotemporal polysaccharide secretion" by Islam et al. describes the genetic determinants of three major polysaccharides secreted by the social bacterium M. xanthus. These three polysaccharides are produced by three different Wzx/Wzy-dependent pathways and expressed differentially through time and space. One of these polysaccharides, BPS, has not been previously described. Its genetic basis and its chemical composition are determined in detail, as wells as the potential implications for sociality in M. xanthus.

This manuscript is very complete in the characterization of a novel polysaccharide. The work is beautifully performed, the paper is very easy to read, clearly written and the figures are exhaustive and illustrate the results precisely. There is a nice balance between genetics, chemistry and social phenotypes. The authors should be commended for this.

I strongly believe that this study is of great quality and, despite the fact that in its current state it may be of interest to a narrow readership (microbiologists working with M. xanthus) it should be accepted for publication with minor changes (see comments below).

Comments

1. I would tone down the claims (abstract and discussion) that this is the first report of an EPS synthesized by a Wzx/Wzy pathway with biosurfactant properties. There have been some capsular polysaccharides (synthesized by the Wzx/Wzy pathway) described to inhibit the formation biofilm (see review  https://doi.org/10.1111/j.1462-2920.2012.02810.x) due to their surface-active properties.

2. I found extremely interesting the fact that the genetic basis for the three polysaccharides is broken up and dispersed in the M. xanthus genome, specially, when in most genomes, Wzx/Wzy-dependent EPS genes are grouped in a contiguous gene cluster, as elegantly shown in Figure 1B. I believe the manuscript would gain relevance for evolutionary biologists if this were a little bit more discussed. Do the authors have any theory? Are there insertions found breaking these gene clusters flanked by mobile genetic elements like transposases, prophages, etc..? Is the GC content of these genes more AT-rich than the rest of the genomes, suggesting that they have been acquired by horizontal-gene transfer?

3. I believe that some measure of similarity between the identified proteins in M. xanthus and other genome(s) should be reported, especially since the cut-off used by blastp is, in my opinion, fairly low (35% query coverage and 35% similarity). I would not call this "stringent" (L510).

4. Concerning the complementation in swarming between BPS- and EPS-, I am a little bit confused with the authors conclusion that BPS is not a shared good, when (i) it is secreted to the environment, and (ii) 10% of BPS+ cells in a mix can complement swarming deficiency of the group. Further, the authors show different combinations of 90:10 BPS-:EPS- and viceversa, but do not comment on the results. It may not be clearer to the reader why they performed this experiment and how to interpret its results. In their opinion, why does 90:10 BPS-:EPS- fully complement swarming, but not 10:90 BPS-:EPS-? This has to be discussed further.

Reviewer #2:

This manuscript describes the genetic organization of genes involved in the synthesis and transport of three different polysaccharides produced by Myxococcus xanthus through distinct Wzx/Wzy pathways. Islam, Alvarez, Saidi et al then focused their works on the characterization of one of these pathways, the BSP pathway, which is required for proper swarming. They analyzed its chemical structure and demonstrated that BSP is a secreted polysaccharide with biosurfactant properties. The authors also showed that single mutants defective in BPS or the cell surface polysaccharide EPS can complement each other in trans when present in a mixed population. Furthermore, they showed that these mutants organize differently in space within these communities and that expression of promoters for EPS and BSP genes are also spatially and temporarily distinct.

The work presented here is solid and well done, and the main conclusions are justified. Overall, the manuscript is also well written. Although I did not find major flaws with the work, I believe that this manuscript would be better suited for a specialized journal, as, in my opinion, it does not meet the description of PLOS Biology publications being of exceptional significance, originality, and relevance. 

Below are some minor points I suggest the authors address to improve their manuscript: 

1. The authors highlight (in the Abstract and the Discussion) the fact that the biosurfactant polysaccharide described here (BSP) is produced by a Wzx/Wzy pathway, but they failed to convey why a reader should find a link between a Wzx/Wzy pathway and a surfactant polysaccharide so novel and interesting. Moreover, I am confused about the fact that they state that this link is novel because, in the Discussion, they refer to a biosurfactant polysaccharide that Acinetobacter makes via a Wzx/Wzy pathway. Please clarify if BSP is indeed the first described biosurfactant that is produced by a Wzx/Wzy pathway and why readers should care about this fact. 

2. Lines 200-204: The authors state: "The only mutants that showed slightly divergent motility and developmental phenotypes compared to other respective EPS- and BPS-pathway mutants were ΔwzxX and ΔwzxB (Fig 2); this is consistent with wzx mutations in one pathway having the potential to affect the biosynthesis of polysaccharides from unrelated pathways (also requiring UndPP-linked precursors) due to depletion of available UndP." To what differences in phenotypes are they referring? The difference in Fig. 2B? None of the other panels show anything of significance, at least to the non-Myxo experts like me. In fact, I think that the authors should consider removing their statement because they do not report the same for the ΔwzyB mutant, which should behave similarly to the ΔwzxB mutant since it would also lead to UndP depletion. I therefore suggest the authors remove these sentences, unless they clearly describe differences that are significantly distinguishing the ΔwzxX and ΔwzxB from the rest of mutants. In contrast, I agree with their argument to explain the differences for wzxB and wzyB mutants in Fig 3A (lines 219-225).

3. Lines 228-231: The authors conclude that "the effect of BPS may be downstream to that of EPS" because the dye-binding profile of the EPS- and BPS-pathway double mutant ΔwzaX ΔwzaB matched that of the EPS pathway ΔwzaX single-mutant. These genetic results indicate that these are independent pathways. They do not imply order.

4. Fig. 3C: Why are the two slopes of the EPS- MASC- double mutant so different? Is that mutant stable or the assay so variable?

5. Lines 386-387: The authors state that BPS is a T4P-regulated polysaccharide. There is no evidence that the effect of T4P on emulsion clearance (Fig. 3E) is occurring through a regulatory mechanism. They should rephrase their statement.

6. Fig. S2B: If EPS mutants are severely defective in swarming motility, what drives their invasion into the E. coli colony? The residual swarming T4P motility or some other type of motility?

Reviewer #3:

[identifies himself as Ákos T. Kovács]

It is a very well written manuscript that provides an insight into polysaccharide production in Myxococccus xanthus, identifying a biosurfactant polysaccahridemin addition to the already know other two apparatuses involved in polysaccharide production. Analyzing mutants reveals that disruption of genes related to BPS production alters spreading motility/swarming, displays aggregative phenotype, and indirect assays suggest that BPS is not cell surface associated unlike EPS. The structure of BPS is further revealed, and the spatial production of EPS and BSP are described in swarming colonies. This is a great contribution to the field of bacterial surface motility, even outside the Myxococcus area. 

The conclusions from the experiments described in L337-L353 (i.e. EPS is public, while BPS private good) are not consistent with the previous observations described in the manuscript, namely, EPS seems to be cell bound, while BPS is not. This can be simply tested by supplying the BPS- strain with BPS overproducing supernatant (DwzaB OpilA strain). In addition, the lack of BPS- strain surface expansion could be also due to hyper-aggregating phenotype, described earlier. Thus, the conclusions from these experiments should be smoothen, and alternative hypothesis also mentioned.

Minor:

L174: at this point of the manuscript, the functionality of the third apparatus is not proven, thus correct: "clusters all encode respective Wzx, Wzy, Wzc, and Wza protein homologues" 

L374: I assume the authors used flow cytometer technique to detect fluorescence (as stated correctly in the methods) and did not actually sort the cells (independently using an instrument with sorting ability)

---

## [Editor Report · Decision Letter 2]

21 May 2020

Dear Dr Islam,

On behalf of my colleagues and the Academic Editor, Tobias Bollenbach, I am pleased to inform you that we will be delighted to publish your Research Article in PLOS Biology. 

Early Version

PRESS 

Kind regards,

Alice Musson

Publishing Editor, 

PLOS Biology

on behalf of

Roland Roberts,

Senior Editor

PLOS Biology